# Differences in walking access to healthcare facilities between formal and informal areas in 19 sub-Saharan African cities

John Friesen [1,2] ✉, Stefanos Georganos [3] & Jan Haas [3]

## Abstract

**Background** Spatial accessibility to healthcare is a critical factor in ensuring equitable health outcomes. While studies on a global, continental, and national level exist, our understanding of intra-urban differences, particularly between formal and informal areas within cities in sub-Saharan Africa, remains limited.

**Methods** This study integrates openly available datasets on land use in 19 sub-Saharan cities, healthcare facilities in the region, and street networks from OpenStreetMap. Using these datasets, we calculate service areas around hospitals, considering travel times ranging from 1 to 120 minutes with walking as the mode of travel. The resulting service areas are then merged with population data from WorldPop, allowing us to assess the proportion of the population with specific travel times to healthcare facilities from informal and formal residential areas.

**Results** Our analysis reveals that 33% of the urban population can reach hospitals within 15 minutes, 58% within 30 minutes, and 78% within 60 minutes. Importantly, for some cities, we observe significant differences between formal and informal areas, with informal areas experiencing a disadvantage in terms of spatial accessibility to healthcare facilities. The population in informal areas is particularly disadvantaged in medium-sized cities.

**Conclusions** This study sheds light on the spatial accessibility of healthcare facilities in sub-Saharan African cities, emphasizing the need to consider intra-urban disparities, particularly in informal areas. The findings underscore the importance of targeted interventions and urban planning strategies to address these disparities and ensure that healthcare services are accessible to all segments of the urban population.

## Plain language summary

This study examines how easily people in 19 sub-Saharan African cities can reach healthcare facilities, focusing on differences between formal and informal neighborhoods. Formal neighborhoods have planned layouts and structured buildings, often linked to higher socio-economic status. Informal neighborhoods have irregular layouts and densely packed buildings, typically associated with lower socio-economic status. Access to healthcare can significantly impact health, yet little is known about how it varies within cities. By mapping walking times to hospitals, we found that 78% of the urban population can reach a hospital within an hour, but residents in informal areas, especially in medium-sized cities, often face longer travel times. This highlights the need for equitable urban planning to improve healthcare access.

Spatial accessibility (SA) to healthcare services is crucial for human well-being and is a necessary prerequisite for achieving universal health coverage, as outlined in target 3.8 of the United Nations Sustainable Development Goals (SDGs). Several studies have investigated accessibility to healthcare services by developing national[1], continental[2], or global maps[3] through travel time analyses. For example, ref. 3 estimated that 56% of the global population can reach a hospital within 1 h. These maps, and the calculated share of the population with certain travel times, provide decision-makers

and local stakeholders with valuable information on the state of healthcare accessibility by providing essential data for effective resource management and spatial planning.

One of the main outcomes of these studies is the expected finding that urban areas typically have a higher coverage of healthcare facilities, with accessibility closely correlating with the degree of urbanization[4]. While these global analyses provide valuable insights, understanding SA within urban environments is particularly critical, especially in rapidly urbanizing regions

[1]Chair of Fluid Systems, Technical University of Darmstadt, Otto-Berndt Str. 2, Darmstadt, 64287, Germany. [2]Earth Observation Research Cluster, University of Würzburg, John-Skilton-Str. 4a, Würzburg, 97074, Germany. [3]Department of Environmental and Life Sciences, Karlstad University, Universitetsgatan 2, Karlstad, 65188, Sweden. ✉e-mail: john.friesen@tu-darmstadt.de; John.friesen@uni-wuerzburg.de

where stark contrasts between urban and rural healthcare accessibility are evident. However, it is also well known that resources and infrastructure are not distributed equally even within cities—a pattern that holds true not only for access to water, sanitation, and energy[5], but also for healthcare[6].

Of the roughly four billion people living in cities today, it is estimated that more than one billion live in deprived urban areas (also called "slums" or "informal settlements"[7–9]). These settlements often lack access to basic amenities such as adequate healthcare services, electricity, and sanitation[10]. Deprived urban areas can be delineated from Earth Observation data, such as satellite images, due to their unique building and settlement morphology[11,12]. Recent studies have pointed out that the communities living in these areas form a particular social group[13], and the United Nations aim to reduce the share of the population living in such areas, as specifically addressed in SDG 11.1.

Nevertheless, few studies investigate the intra-urban differences in SA to healthcare. Until recently, these were mainly limited to cities in Latin America[14] and Southeast Asia[15]. Banke-Thomas et al.[16] took a first step toward intra-urban analysis of access to maternity care in SSA by analyzing travel times for 15 Nigerian cities. They found that informal settlements appear to be worse off compared to parts of the inner city. Dumedah et al.[17] analyzed access to healthcare in four Ghanaian cities. Similar to the studies mentioned above, they found that relative wealth negatively correlates with access to healthcare.

Although the last two studies compared different parts of the respective cities, there is still a systematic lack of studies comparing SA to healthcare facilities, taking into consideration the following criteria: (i) high geographic variability concerning the number of countries investigated and (ii) capturing variations in urban morphology. Until recently, there was a lack of large-scale and spatially detailed datasets on urban land use, making it extremely challenging to tackle such objectives rigorously.

It has already been widely studied that residents of informal settlements are exposed to a variety of health risks, such as an increased likelihood of contracting communicable diseases[7,18]. Systematic studies on whether and to what extent this population group is disadvantaged in terms of spatial access to healthcare facilities are still missing[19,20]. At the same time, it is known that longer travel times to healthcare facilities are a risk factor for more severe disease progression, such as in acute cardiovascular diseases, and can favor complications in delivery[21,22].

Our analysis employs cutting-edge spatial datasets to model walking accessibility to healthcare facilities across 19 cities in sub-Saharan Africa, considering urban morphology as a proxy for social groupings.

With this approach, we aim to broaden the understanding of accessibility to healthcare in deprived urban areas by using recently produced datasets of settlement types from the World Resource Institute, which classifies land use in more than 200 cities worldwide, 19 of which are in sub-Saharan Africa (SSA)[23].

We combine this dataset with geospatial information on healthcare facilities in SSA[24], data on road networks provided by OpenStreetMap (OSM), and WorldPop, a commonly used and recommended gridded population dataset[25]. This enables us to calculate walking accessibility to healthcare in these SSA cities for travel times between 0 and 120 min.

Leveraging novel datasets and geospatial analysis techniques, this study seeks to advance the understanding of healthcare accessibility in deprived urban areas of sub-Saharan Africa. We specifically aim to address the following research questions:

(i) How good is the walking accessibility to healthcare facilities in different cities in SSA, and how does this compare to global or national estimates? (ii) Are there differences in walking accessibility between people living in formal and informal urban areas? (iii) Is living in an informal area a risk factor for having limited access to healthcare?

Our findings reveal that 78% of the urban population across 19 sub-Saharan African cities can reach a healthcare facility within 60 min of walking. However, disparities emerge when comparing formal and informal neighborhoods, with residents in informal areas, especially in medium-sized cities, often experiencing longer travel times. The relative risk analysis shows

that living in an informal area is associated with significantly worse healthcare accessibility in some cities. These results highlight the critical role of urban morphology and city size in shaping healthcare access, underscoring the need for targeted interventions and urban planning to reduce these disparities.

## Methods
### Theoretical background
Access to a place or service can be assessed along a number of dimensions, as outlined in a comprehensive theory by ref. 26. This theory begins with (i) where to measure access, (ii) what kind of barrier or cost to consider, (iii) how to assess access, (iv) what time of day to assess access, (v) what place to assess access, (vi) what mode of transport to consider, and (vii) what group of people to consider[26].

We applied aspects of Levinson and Wu's theory to define healthcare access in our study, focusing on (i) urban areas in SSA and (ii) travel time via (iii) isochrones. Due to data scarcity, we (iv) did not account for daily fluctuations. The study focused on (v) healthcare facilities, using (vi) walking as the travel mode, and (vii) comparing urban groups based on urban morphology, which serves as a proxy for living conditions shown to correlate with social groups[13].

### Data sources
Accessibility was modeled by calculating the travel time to the closest healthcare facility through transportation network analysis. The analysis was performed in ESRI ArcGIS Pro 3.2.1 using the Network Analyst extension. The network was built using OSM data which is freely available.

We calculated service areas for health facilities in 19 cities, representing all cities from a previous analysis[23] that are located in SSA. Additionally, we performed an analysis focusing solely on hospitals to compare the results of our methodology with similar studies[2,16]. For detailed information on the classification of both groups of healthcare facilities, see below.

The mode of travel was defined as walking, as it is reported to be the main mode of transportation in SSA[27] and is independent of income. Additionally, it is reported that, for example, in Kenya, more than 80% of patients access healthcare by walking[28], and the corresponding travel time can be seen as a lower bound. Similar to other studies analyzing walking travel times to healthcare facilities[28], we set the walking speed to 3.6 km/h. The OSM road network allows free traversal but mainly maps motorized roads, lacking data on footpaths or shortcuts that could reduce pedestrian travel time. Additionally, OSM data do not provide systematically detailed road classifications, conditions, or other factors affecting walking speed, but main roads are often used as proxies for pedestrian paths[28].

To account for intra-urban variations in walking speed (e.g., due to topography) and to test the robustness of the results, we varied the speed by ±20%, similar to ref. 2 and ref. 28.

Since some of the health facilities are outside of the city boundaries, a 9-km buffer was chosen as it represents the maximum rounded distance that an individual could theoretically reach within 2 h on foot, based on a walking speed of 3.6 km/h with a 20% uncertainty margin. This ensures that all healthcare facilities within a reasonable travel distance, including those just outside city boundaries, are considered in the accessibility analysis.

We opted to use street networks instead of friction surfaces (like ref. 2) to calculate travel time because they provide a more accurate reflection of pedestrian movement within urban environments. Street networks capture the actual routes people take, considering the layout and connectivity of roads, which is crucial in dense urban areas, especially where informal settlements are prevalent. While friction surfaces offer a broad approximation, they often lack the precision needed to represent the complexities of urban movement. Using street networks ensures a more realistic assessment of healthcare accessibility in SSA.

For each one-minute time step up to 2 h, 120 service area polygons (isochrones) around healthcare facilities were created. Overlapping service areas were dissolved and combined into fewer but larger areas. We used freely available data on healthcare facilities from ref. 24, who collected

information on 98,745 geocoded public health facilities in SSA until 2018 from ministries of health, state agencies, United Nations Coordination of Humanitarian Affairs, and other sources. Further information on the dataset can be found in the data descriptor paper[24]. All 19 investigated cities are covered in this dataset. Since the healthcare datasets contain information on various healthcare facilities, from dispensaries and health clinics to national hospitals, we categorized the data according to their specialization. We divided them into two groups according to the table provided in Supplementary Data 5. The first group (i) refers to hospitals and centers with surgical care, while the second group (ii) includes health clinics and other small health facilities.

In our analysis, we focused on all healthcare facilities (group (ii)). To be able to compare our findings with other studies on SA, we performed an additional analysis using just a subset of the data (group (i)).

### Population and morphology analysis

As mentioned previously, we rely on urban morphology as a proxy for the social groups living in the respective parts of the city. Urban morphology, which refers to the physical form and structure of urban environments, is a well-established approach in urban studies to distinguish between different types of settlements[29]. This approach is particularly useful when direct socio-economic data are unavailable, difficult to obtain, inconsistent across different countries, or have lower spatial resolutions (e.g., data from demographic health surveys). In this study, urban morphology is leveraged to differentiate between formal and informal urban settlements, as the physical characteristics of these areas—such as building density, layout, and infrastructure—often reflect underlying socio-economic conditions[13].

To distinguish between different types of land use within city boundaries, we used the classification of ref. 23. Their freely available classification is based on a machine learning algorithm with a resolution of around 5 m and has classified more than 200 cities worldwide. The group aims to classify all cities worldwide with more than 100,000 inhabitants, and the published dataset is the first subset with exemplary cities in different countries and of various sizes. They performed several land use and land cover classifications at different levels. We used the categorization that includes six land use classes (atomistic, housing projects, formal land subdivision, informal land subdivision, open space, and non-residential). Guzder Williams et al.[23] provide the following definitions of the different residential categories: Atomistic settlements are "areas occupied for settlement before planning had occurred." Informal land subdivisions refer to "areas occupied for settlement and presumed planned informally, based on visible infrastructure quality, parcel sizes, road widths, and connection to arterial roads." Formal land subdivisions are "areas occupied for settlement and presumed planned formally, with approval of the municipal government, based on visible infrastructure quality, parcel sizes, road widths, and connections to arterial roads." Finally, housing projects are "areas occupied for settlement with home construction and land subdivision conducted under the same plan, based on layout and similarity of structures"[23]. As a final spatial land use layer, we combined "atomistic" and "informal subdivisions" into a new category named "informal land use." For "formal land use," we merged the sub-classes "housing projects" and "formal subdivision." This procedure was also performed in the original paper to assess the dataset's ability to detect informal settlements[23].

The distinction between formal and informal settlements is critical to understanding urban inequalities. Formal settlements are generally planned and regulated, with better access to services and infrastructure, whereas informal settlements often emerge spontaneously and lack such resources. These differences in urban form are not just physical; they are deeply intertwined with socio-economic factors such as income levels, employment types, and access to public services[9]. Nonetheless, we justify the use of urban morphology in this study due to the lack of comprehensive socio-economic data at the necessary spatial scale across the cities we examined. The classification by ref. 23 provides a standardized, globally applicable framework that allows for consistent comparison across diverse urban contexts. This approach is particularly useful in rapidly urbanizing regions where data

collection can be challenging, and where urban form often remains a strong indicator of different social groups[13]. We used information from all 19 sub-Saharan cities available.

By calculating the intersection between the service areas around healthcare facilities and the two settlement types, we were able to show areal differences in accessibility to healthcare.

To calculate the population living within the service areas, we used the freely available WorldPop Constrained UN adjusted dataset[25] for the year 2020 with a spatial resolution of around 100 m. We based our choice on the comparative study of ref. 30, who compared different population datasets in a similar study. Although the datasets used here can be highly uncertain, especially in the area of informal settlements (cf. refs. 31,32), they are nonetheless the best datasets available on a continental scale and thus make comparative studies possible in the first place. We assumed a uniform distribution within the gridded population cells and dis-aggregated the cells into a 5-m population grid to fit the population data into the land use layer. Ultimately, by merging the population layer with the service areas for the respective travel times and land use, we conducted a comprehensive SA analysis for each city. SA is defined as the share of the total population (cf. Table 1) with travel times to a healthcare facility below a certain threshold. Here, we distinguish between the population of the city as a whole and the population within formal and informal areas of the city.

By setting the SA for both land use types in relation, we calculated the relative risk $RR = SA_F/SA_{IF}$ (Formal $F$ and Informal $IF$) for different steps in time. The relative risk (RR) quantifies the disparity in healthcare accessibility between formal and informal areas. An RR greater than 1 indicates higher accessibility in formal areas, while an RR less than 1 suggests greater accessibility in informal areas.

We analyzed the accessibility to healthcare facilities for 120 1-min intervals, with emphasis on 15-, 30-, and 60-min breakpoints. Besides these values mentioned in the literature (e.g., ref. 3), we also assessed the extent to which different cities meet the recommendation that 80% of the population should reach surgical care within 2 h[33]. For this analysis, we focused solely on hospitals (Group (i)) and compared our findings with those of ref. 2.

To summarize, we fused different datasets: (i) land use/land cover (LULC)[23], (ii) residential population[25], (iii) street networks from OSM, and (iv) healthcare facilities[24] to calculate walking accessibility to healthcare infrastructure. While we recognize several limitations inherent in the datasets used, which we discuss below, we emphasize that our study utilizes the best available data to contribute valuable insights into healthcare accessibility challenges in SSA.

This study did not require IRB approval as it does not involve human participants, interventions, or personally identifiable information. The datasets used are publicly available, and the modeled nature of the population and settlement data ensures no direct involvement of human subjects.

### Reporting summary

Further information on research design is available in the Nature Portfolio Reporting Summary linked to this article.

## Results

We analyzed 19 cities in SSA. When merging the population data from WorldPop with both land uses (formal and informal), all cities combined have an estimated total population of 56.9 million (Table 1). Categorizing the cities according to the city size classification of UN Habitat[10], we included four very large cities with more than 5 million inhabitants (Johannesburg, Kinshasa, Lagos, and Luanda), seven large cities with a population between 1 and 5 million inhabitants (Accra, Addis Ababa, Bamako, Ibadan, Kampala, Luanda, Lubumbashi), six medium-sized cities with a population between 250,000 and 1 million inhabitants (Arusha, Beira, Kigali, Ndola, Oyo, and Port Elizabeth), and two small cities (Gombe and Nakuru).

With the exception of Addis Ababa, Johannesburg, and Port Elizabeth, the proportion of the population living in informal areas is above 50%, and

**Table 1 | Cities investigated in this study**

| City | Total pop. in 2020 in mio. | City category according UN | Percentage of population living in informal areas | Number of healthcare facilities / thereof hospitals |
|---|---|---|---|---|
| Accra, Ghana | 3.17 | large | 90.7% | 92/12 |
| Addis Ababa, Ethopia | 2.82 | large | 14.7% | 460/37 |
| Arusha, Tanzania | 0.52 | medium-sized | 43.6% | 135/34 |
| Bamako, Mali | 3.85 | large | 84.7% | 215/28 |
| Beira, Mozambique | 0.40 | medium-sized | 94.9% | 30/1 |
| Gombe, Nigeria | 0.24 | small | 96.9% | 218/12 |
| Ibadan, Nigeria | 2.19 | large | 97.2% | 332/16 |
| Johannesburg, South Africa | 8.22 | very large | 10.0% | 360/21 |
| Kampala, Uganda | 2.88 | large | 72.1% | 205/33 |
| Khartoum, Sudan | 3.41 | large | 98.6% | 40/40 |
| Kigali, Ruanda | 0.84 | medium-sized | 80.0% | 94/10 |
| Kinshasa, Democratic Republic Congo | 5.19 | very large | 97.3% | 1337/7 |
| Lagos, Nigeria | 8.24 | very large | 95.4% | 265/21 |
| Luanda, Angola | 10.94 | very large | 77.8% | 10/9 |
| Lubumbashi, Demcratic Republic Congo | 2.27 | large | 94.3% | 1003/1 |
| Nakuru, Kenya | 0.15 | small | 65.6% | 72/4 |
| Ndola, Zambia | 0.33 | medium-sized | 70.7% | 79/0 |
| Oyo, Nigeria | 0.28 | medium-sized | 93.4% | 153/10 |
| Port Elizabeth, South Africa | 0.99 | medium-sized | 33.9% | 71/4 |
| Total | 56.9 | | 71.9% | 5171 |

The total population was calculated by merging WorldPop data with the land use data from ref. 23. The healthcare data was taken from ref. 24 and includes all healthcare facilities inside the abovementioned 15-km buffer zone around the land use maps of each city.

in nine cities, it is above 90%. Taking all cities together, we find that 71.9% of the total population lives in informal areas.

We also observe substantial differences in the number of healthcare facilities relative to city size, as provided in the dataset of ref. 24. While some smaller cities have a high number of registered healthcare facilities (for example, Oyo has 153 healthcare facilities per 280,000 inhabitants), other larger cities such as Luanda, have only ten registered healthcare facilities. To what degree this variation can be attributed to factual inter-city differences or incomplete survey data cannot be quantified and will be discussed later on.

**Accessibility to healthcare facilities**
Analyzing access to healthcare facilities by walking in all 19 SSA cities combined, we find that 18.6 million, or 32.7% [26.3–38.7], of the urban population can reach a healthcare facility within 15 min. Almost 58% [49.3–64.1] of the urban population can reach a healthcare facility within 30 min, and 78% [73.0–82.6] can reach one within 60 min.

A comparison of accessibility among the investigated cities (Fig. 1) reveals that the share of population with travel times below 15 min to the next healthcare facility ranges between 3% in Luanda and 90% in Kinshasa with a median value of 30% for all cities. When considering travel times below 30 min, the share of the population varies between 13% in Luanda and 98% in Kinshasa with a median of 65%. The share of the population with travel times below 60 min varies between 32% for Luanda and 99% in Kinshasa with a median of 94%.

Substantial discrepancies become apparent when the development of SA is considered for the different cities (Fig. 2). While some cities reach high SA values for short travel times (below 20 min) (Kinshasa, Kigali, and Addis Ababa), (very) large cities like Luanda or Khartoum show low SA for all travel times.

Although accessibility, as expected, reaches saturation for long travel times, the SA curves in Fig. 2 differ, however, with few intermediate plateaus. This can be interpreted as an indicator that accessibility is a function of local topography and morphology. Plateaus like the one for Beira emerge due to the urban topology with different urban centers represented in the dataset. The development of SA to hospitals is shown in Fig. S2 of Supplementary Information.

**Differences in accessibility based on residential land use**
Apart from these general observations, we analyze how walking accessibility to healthcare differs between formal and informal settlements. We find that SA is lower in informal areas than in formal areas. Looking at all cities combined, we find that 32.2% [26.3–37.8] of the population in informal areas have access to healthcare within 15 min, compared to 34.1% [26.3–41.3] in formal areas. We find that 55.4% [47.6–61.8] of the population in informal areas and 63.2% [53.7–70.2] in formal areas can reach healthcare facilities by walking 30 min or less, and that 76.3% [70.8–80.4] of the population in informal areas and 84.4% [78.8–88.1] in formal areas have travel times below 60 min.

Looking at each city in turn (Fig. 3), we find a wide range of SA for both informal and formal housing when considering travel time intervals of 15 and 30 min. SA in informal areas within 15 min ranges from 2% in Luanda to 90% in Kinshasa and from 5% in Accra to 88% in Lubumbashi for formal areas. When looking at SA for 30 min, we observe accessibility between 13% and 98% in informal areas and between 20% and 96% in formal areas.

Table 2 shows the relative risks RR for travel time intervals of 15, 30, and 60 min. Values significantly higher than 1 indicate that living in an informal area increases the risk of having limited access to health facilities.

We find that the population in informal areas of a city has significantly worse access to healthcare facilities for all travel times (15, 30, and 60 min)

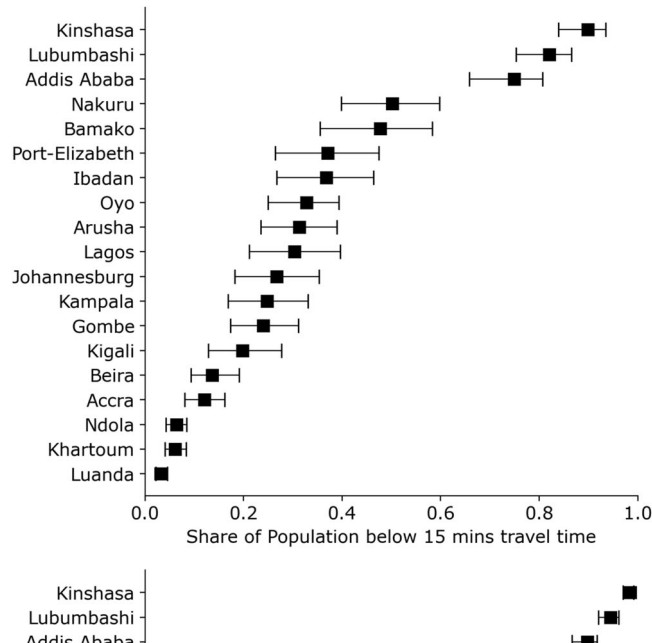

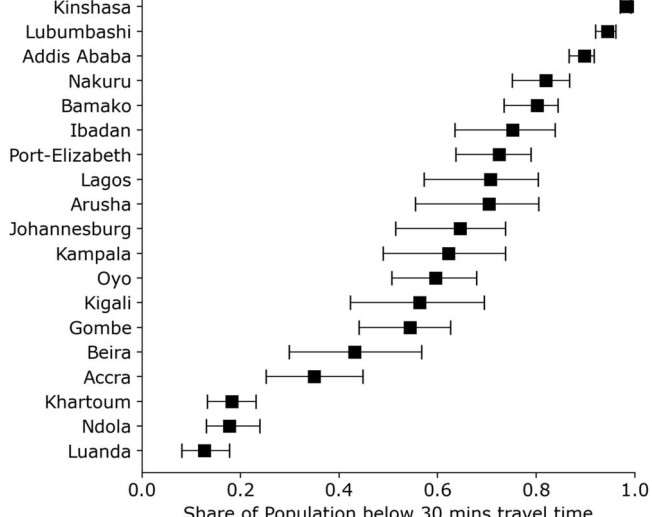

**Fig. 1 | Share of population with defined travel times.** Modeled spatial accessibility to healthcare facilities (Group (ii)) for all cities (both land uses combined) for 15 and 30 min travel time. Error bars represent a modeled varying walking speed of ±20%. The source data for Fig. 1 is in Supplementary Data 1.

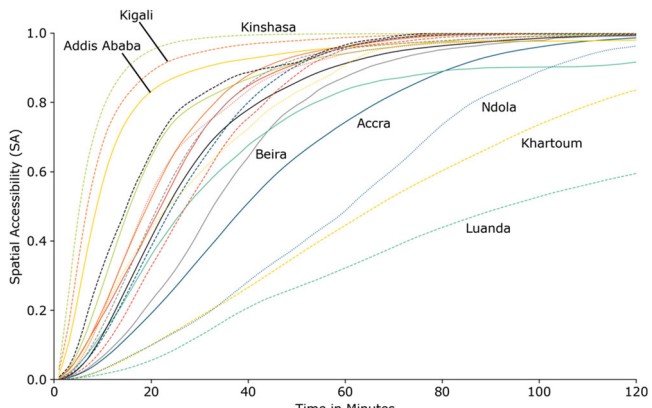

**Fig. 2 | Share of population for all travel times (1–120 min).** Modeled spatial accessibility to all healthcare facilities (hospitals, clinics, and smaller health facilities - Group (ii)) for all (*n* = 19) full urban areas for travel times between 1 and 120 min. A similar figure with the modeled uncertainties included a full description of all cities is shown in Fig. S1 of Supplementary Information. The source data for Fig. 2 is in Supplementary Data 2.

than the population in formal areas in two large cities (Addis Ababa and Khartoum) and two medium-sized cities (Beira and Ndola). In four cities (Arusha, Bamako, Luanda, and Oyo), RR is significantly high only for one or two of the travel times, where the uncertainty limits for the exceptions are not higher than 1.

The highest values can be observed for Khartoum, with RR values of 5.98, 4.07, and 2.04 for 15, 30, and 60 min, respectively. This means that individuals living in informal areas in Khartoum are almost six times more likely to have a travel time higher than 15 min compared to individuals living in formal areas of the city. It is noteworthy that cities with higher spatial accessibility, such as Addis Ababa, cities with average SA (Fig. 1), such as Arusha or Oyo, as well as cities with poor spatial accessibility (Khartoum, Luanda, or Ndola), can be found in this list. Therefore, the relative risk can be considered independent of SA. Individuals living in informal areas within these cities are more likely to have longer travel times to healthcare facilities compared to urban dwellers in the formal parts of the respective cities.

### Influence of city size
The relative risk presented in Table 2 is shown in Fig. 4 with additional information on the population of the respective cities. We observe a trend of *RR* toward 1 for higher travel times. Another finding is that, for large and very large cities, an irregular pattern can be observed. While the *RR* for some cities is below 1 and increases with travel time, other cities exceed this threshold while showing a decreasing trend with increasing travel time.

Looking at small cities, we find that the *RR* is below or equal to 1, while the *RR* for medium-sized cities is above 1. For the city sample, we investigated in this study, we find that individuals in informal areas in medium-sized cities are disadvantaged when it comes to accessibility to healthcare facilities. This finding can be explained by the mismatch in medium-sized cities between rapid urban growth (often exceeding that of primary cities) and the provision of resources in infrastructure and healthcare[34,35].

### Maps of travel times
In Fig. 5, the service areas for 30 and 60 min are shown for formal and informal areas within the investigated cities. The porous structure of some cities (such as Addis Ababa, Johannesburg, or Kampala) refers to the fact that large parts of these cities are classified as non-residential, and therefore no population is assigned to these areas.

Confirming and visualizing the results of Fig. 3, it can be seen that almost every investigated city has formal and informal parts where the travel time to the nearest healthcare facility is higher than 30 and 60 min, indicated by bright blue or yellow areas. The high percentage of unserved areas is particularly apparent in Khartoum, Luanda, and Ndola.

It is evident that certain cities, notably Addis Ababa, Arusha, Johannesburg, and Port Elizabeth, not only exhibit a relatively low percentage of their population residing in informal areas (refer to Table 1), but also feature a lower proportion of informal areas within the city compared to other cities.

Moreover, we find across many cities, including Addis Ababa, Arusha, Kampala, Lubumbashi, Ndola, and Port Elizabeth, that formal areas are predominantly situated in central locations, while informal areas tend to be more peripheral. Additionally, there is a discernible pattern of healthcare facilities being centrally located within the city. This aligns with the observed high relative risk for individuals from informal areas, indicating a greater likelihood of encountering poor access to health facilities.

### Discussion
Our analysis reveals significant differences in spatial accessibility (SA) to healthcare facilities across various sub-Saharan African (SSA) cities, underscoring the intricate relationship between urban planning, land use, and health equity. The disparities in SA between formal and informal areas within these cities highlight the ongoing challenges in urban health provisioning, particularly in rapidly expanding urban environments. These findings are critical as they emphasize the uneven distribution of healthcare resources, which can have profound implications on public

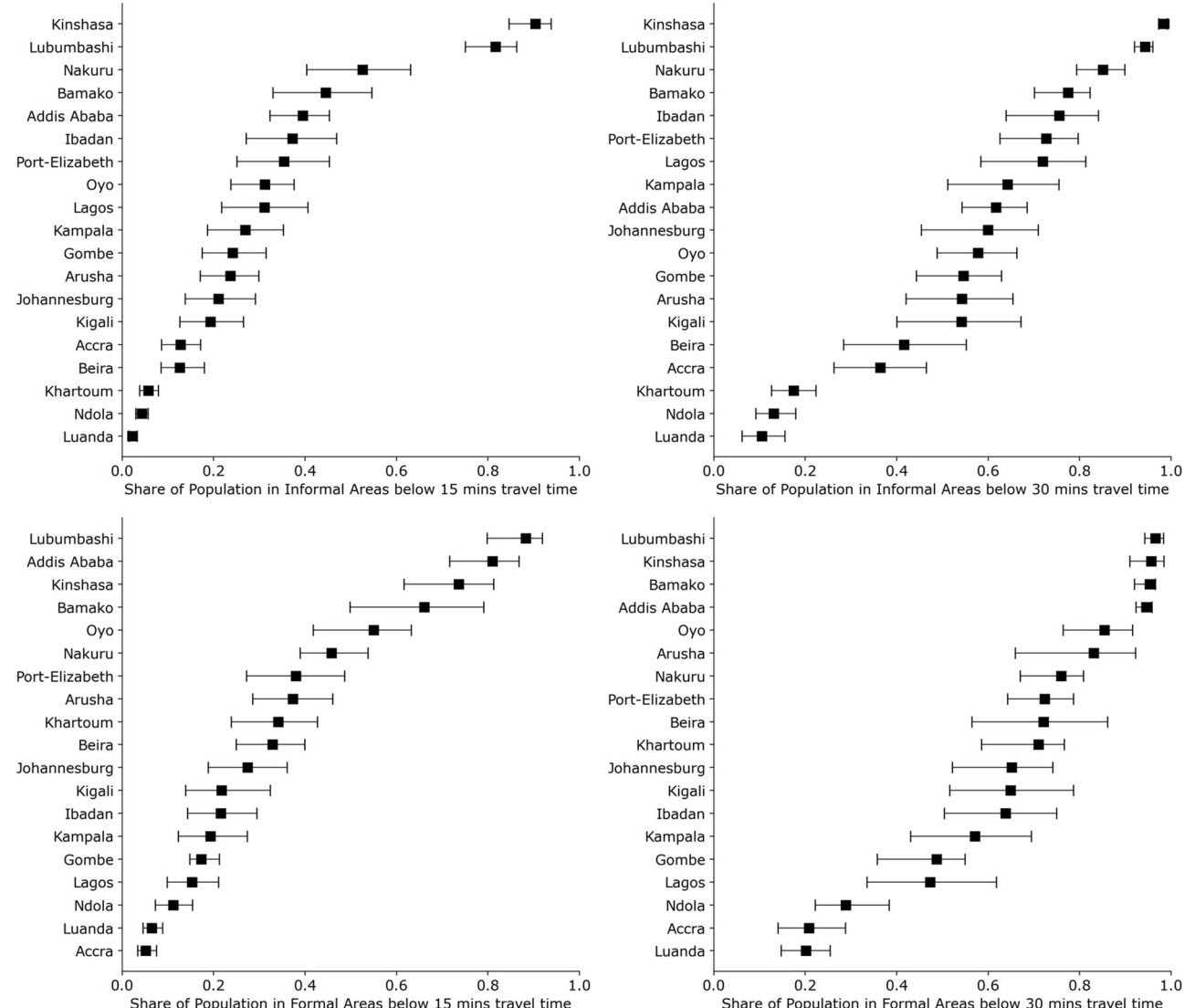

**Fig. 3 | Share of population with defined travel times for different classes of residential land use.** Modeled spatial accessibility (SA) for informal and formal residential areas to healthcare facilities (Group (ii)) for travel times 15 and 30 min.

Error bars represent a modeled varying walking speed of ±20%. The source data for Fig. 3 is in Supplementary Data 3.

health outcomes, especially for vulnerable populations residing in informal settlements.

While ref. 3 found that 39.8 and 56.9% of the global population can reach healthcare by foot within 30 and 60 min, respectively, we show that the values for urban populations are higher. We find that 58 and 78% of the urban population in the SSA cities surveyed can reach a health facility within 30 and 60 min. Nevertheless, we found that the values differ when a distinction between formal and informal areas is made. For example, 63.2% of the population living in formal areas has travel times below 30 min, while only 55.4% of the population in informal areas meets this threshold. This disparity not only reflects the historical neglect of informal settlements in urban planning but also poses a significant public health risk, as these populations may be more vulnerable to outbreaks of disease and other health crises.

Based on previous analyses, we were also able to calculate the relative risk of having access below a certain travel time for the cities considered in this study. We found that in some of the cities (Addis Ababa, Beira, Khartoum, and Ndola), living in an informal area is a risk factor for having limited access to healthcare facilities. Since the relative risk is independent of the SA and the size of the respective city, further work is required to understand the implications of these findings.

Interestingly, our analysis (cf. Fig. 4) confirmed the importance of medium-sized cities, as mentioned in other publications such as the World Cities Report[10] or the sixth report of the IPCC[36]. Both publications highlight the importance of these areas, often referred to as secondary cities, due to their high growth rates in area and population. We emphasize their particular importance by finding that, especially in these cities, populations living in informal areas are disadvantaged in their access to healthcare.

The travel time maps and the heterogeneity of spatial accessibility (SA) to health infrastructures within informal areas highlight the diverse living conditions for inhabitants, even with similar morphology serving as the basis for area identification. Our results, derived from satellite imagery, underscore the variability in living conditions within these settlements. This emphasizes the significance of comprehensive analysis, as advocated by global initiatives like IDEAMaps[9], to effectively implement suitable upgrading concepts. Assuming that the dataset provided by ref. 24 is accurate, these maps provide valuable information on the locations within the city where healthcare infrastructure is necessary.

In our study, we analyzed accessibility to healthcare facilities specifically by walking, whereas ref. 16 assessed accessibility to comprehensive

emergency obstetric care facilities across 15 Nigerian cities using travel time estimates from Google Maps, which primarily include driving as the mode of transportation. This methodological difference allows for a meaningful comparison between the two approaches in the context of Lagos and Ibadan.

For Ibadan, ref. 16 reported spatial accessibility (SA) of 44.9% within 15 min and 79.8% within 30 min when considering only public facilities. In contrast, our study, which focuses solely on hospitals (Group (i)) and considers walking as the mode of travel, found significantly lower SA values of 2.9% for 15 min and 12.6% for 30 min. Similarly, in Lagos, they found SA

values of 45.8% for 15 min and 83.7% for 30 min, while our study found 4% (15 min) and 15.9% (30 min).

These stark differences in SA percentages can be primarily attributed to the mode of transportation considered in each study. By focusing on walking, our approach is inherently more conservative and is designed to detect population groups with insufficient access to healthcare under more constrained mobility conditions. In contrast, ref. 16 use of driving times likely reflects faster travel times and greater accessibility in urban areas. Although the number of hospitals considered was similar between the two studies, the choice of transportation mode plays a crucial role in accessibility outcomes. Therefore, while both sets of results are reasonable within their respective contexts, our conservative approach highlights potential disparities in healthcare access that might be overlooked when considering faster modes of transportation.

With our results, we are also able to compare intra-urban SA with national SA to hospitals (Group (i)) in SSA (cf. Table 3). We observe various patterns. For the cities of Accra, Beira, and Nakuru, we observe that the citywide percentage of the population outside 2-h travel time is above the national values. Nevertheless, while in all of these cities, the percentage of the population outside a 2-hour travel time living in formal areas is below the national average, the value for informal areas is higher. This indicates that the population in informal areas is disadvantaged not only in comparison to formal areas within the city but also in comparison to the city as a whole. These findings are supported by the relative risks presented in Table S1 in Supplementary Information, where seven cities show an increased risk for individuals in informal areas of having worse access to hospitals within 120 min than those living in formal areas.

In contrast, we also identified several cities where national values are considerably higher than citywide formal and informal SA values (Addis Ababa, Arusha, Bamako, Khartoum, Kinshasa, Luanda, Lubumbashi, and Oyo). All of these cities confirm the often observed finding that large cities are often advantaged in comparison to other parts of the country in terms of accessibility to healthcare[3]. Lastly, we find that in Johannesburg and Port Elizabeth, the national value for South Africa is considerably lower than the citywide, informal, and formal values. Given that South Africa is reported to have higher SA to healthcare compared to other SSA cities, this finding needs further investigation and could be influenced by the data used to conduct our study and its associated limitations.

The relevance of this study extends beyond the immediate findings on spatial accessibility (SA) to healthcare facilities in Sub-Saharan African (SSA) cities. The disparities highlighted between formal and informal urban settlements underscore broader issues of inequality that are crucial to urban planning and public health. In rapidly urbanizing regions, where informal settlements continue to expand, ensuring equitable access to essential services like healthcare is a pressing challenge. These findings underscore the need for urban planners and policymakers to prioritize the inclusion of informal areas in urban development plans, particularly in healthcare provisioning.

**Table 2 | Relative risk for having worse access to healthcare (Group (ii)) when living in an informal area compared to living in a formal area for different travel times**

| City | Relative risk 15 min | Relative risk 30 min | Relative risk 60 min |
|---|---|---|---|
| Accra, Ghana | 0.40 [0.20–0.88] | 0.57 [0.30–1.10] | 0.86 [0.53–1.26] |
| Addis Ababa, Ethiopia | **2.05 [1.58–2.69]** | **1.53 [1.35–1.77]** | **1.21 [1.12–1.30]** |
| Arusha, Tanzania | 1.58 [0.96–2.70] | **1.53 [1.01–2.19]** | **1.08 [1.01–1.25]** |
| Bamako, Mali | 1.48 [0.91–2.40] | **1.23 [1.12–1.38]** | **1.06 [1.03–1.12]** |
| Beira, Mozambique | **2.60 [1.39–4.71]** | **1.73 [1.02–3.03]** | **1.12 [1.03–1.29]** |
| Gombe, Nigeria | 0.72 [0.47–1.22] | 0.89 [0.57–1.24] | 0.93 [0.80–1.09] |
| Ibadan, Nigeria | 0.58 [0.30–1.09] | 0.84 [0.60–1.17] | 0.97 [0.90–1.05] |
| Johannesburg, SA | 1.30 [0.65–2.61] | 1.09 [0.74–1.63] | 0.95 [0.86–1.08] |
| Kampala, Uganda | 0.72 [0.35–1.47] | 0.89 [0.57–1.36] | 0.98 [0.87–1.09] |
| Khartoum, Sudan | **5.98 [3.01–11.24]** | **4.07 [2.62–6.09]** | **2.04 [1.54–2.81]** |
| Kigali, Rwanda | 1.13 [0.52–2.58] | 1.20 [0.77–1.97] | 1.04 [0.96–1.18] |
| Kinshasa, DRC | 0.81 [0.66–0.96] | 0.97 [0.92–1.01] | 1.00 [0.99–1.00] |
| Lagos, Nigeria | 0.49 [0.24–0.97] | 0.66 [0.41–1.06] | 0.95 [0.85–1.03] |
| Luanda, Angola | **2.92 [1.38–6.01]** | 1.91 [0.95–4.12] | 1.29 [0.87–1.95] |
| Lubumbashi, DRC | 1.08 [0.92–1.22] | 1.02 [0.98–1.07] | **1.01 [1.00–1.02]** |
| Nakuru, Kenya | 0.87 [0.62–1.33] | 0.89 [0.75–1.02] | 0.94 [0.85–1.05] |
| Ndola, Zambia | **2.58 [1.29–5.12]** | **2.20 [1.24–4.17]** | **1.76 [1.04–2.97]** |
| Oyo, Nigeria | **1.76 [1.11–2.66]** | **1.48 [1.15–1.87]** | 1.05 [0.98–1.24] |
| Port Elizabeth, SA | 1.07 [0.60–1.94] | 1.00 [0.81–1.26] | 0.93 [0.88–1.02] |

Values (including uncertainty) above one are written in bold. The values inside the brackets show the uncertainty due to a possible change in walking speed by 20%.
*SA* South Africa, *DRC* Democratic Republic Congo.

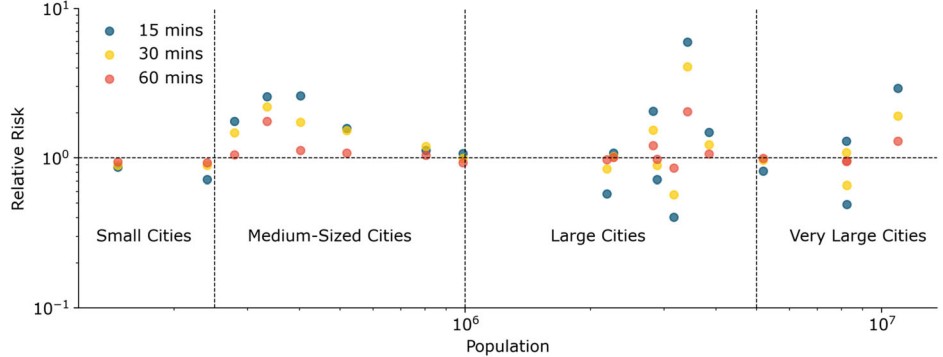

**Fig. 4 | Relative risk of having limited access to healthcare (Group (ii)) for different travel times in dependence of city size.** Dots above the dashed line indicate a relative risk for the population living in informal areas of having reduced access to healthcare in comparison to the population in formal areas. The source data for Fig. 4 is in Supplementary Data 4.

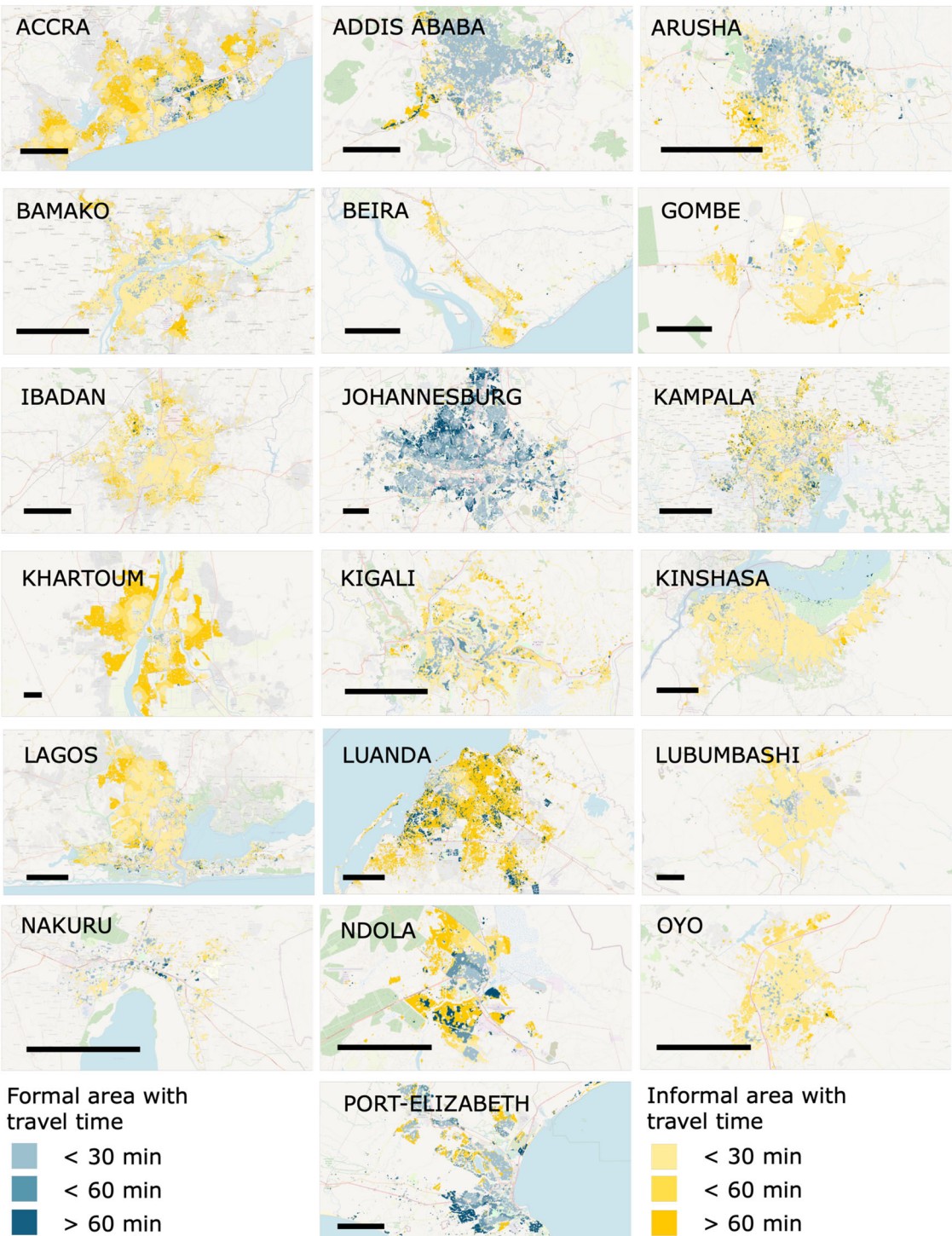

**Fig. 5 | Maps of service areas around healthcare facilities (Group (ii)) for different steps in time.** The bar describes a distance of 10 km.

Urban planning often prioritizes formal areas due to their regulated nature and easier integration into city-wide infrastructure networks. However, as this study demonstrates, neglecting informal settlements can exacerbate existing health inequities, leaving large segments of the urban population vulnerable to disease outbreaks and other health crises. The study's findings serve as a critical reminder that achieving health equity in urban settings requires an inclusive approach that addresses the needs of all urban residents, regardless of the formal status of their living conditions[37].

The disparities in healthcare accessibility revealed by this study have profound implications for public health outcomes in SSA cities. Limited access to healthcare facilities in informal settlements not only increases the risk of untreated illnesses but also contributes to broader public health challenges, such as the spread of infectious diseases. The relative risk analysis indicating that residents of informal settlements are more likely to experience limited access to healthcare reinforces the urgent need for targeted public health interventions in these areas.

To work as consistently as possible, we used globally or continentally available products such as the healthcare dataset from ref. 24, the WorldPop data[25] for gridded population maps, land use data[23] to distinguish between formal and informal areas, and OpenStreetMap to

**Table 3 | Percentage of population outside 2-h travel time to the next hospital (Group (i))**

| City<br>Travel mode | Citywide<br>(walking) | Informal<br>(walking) | Formal<br>(walking) | National<br>(various) |
|---|---|---|---|---|
| Accra, Ghana | 27.6% [21.6–37.1] | 28.9% [23.0–37.8] | 14.8% [7.6–30.1] | 13.8% [12.8–15.5] |
| Addis Ababa, Ethiopia | 6.2% [4.4–10.1] | 21.2% [16.1–34.5] | 3.6% [2.4–6.0] | 49.3% [45.5–54.3] |
| Arusha, Tanzania | 2.1% [0.6–8.3] | 4.3% [1.3–16.8] | 0.3% [0.1–1.8] | 24.9% [22.1–29.0] |
| Bamako, Mali | 12.6% [8.4–18.5] | 14.6% [9.7–21.0] | 1.8% [1.2–4.5] | 36.2% [33.3–40.6] |
| Beira, Mozambique | 59.0% [56.9–64.8] | 60.0% [57.9–65.8] | 41.0% [37.7–47.6] | 49.9% [44.8–52.7] |
| Gombe, Nigeria | 39.4% [26.3–56.1] | 39.6% [26.0–56.6] | 35.5% [34.7–41.8] | 7.7% [6.9–9.0] |
| Ibadan, Nigeria | 20.3% [15.4–26.9] | 20.4% [15.5–27.0] | 16.8% [12.9–23.8] | 7.7% [6.9–9.0] |
| Johannesburg, South Africa | 43.9% [35.0–57.1] | 53.7% [45.7–65.3] | 42.7% [33.8–56.2] | 5.2% [4.4–6.3] |
| Kampala, Uganda | 6.0% [3.2–11.9] | 4.3% [1.9–9.0] | 10.4% [6.5–17.0] | 17.5% [15.5–20.7] |
| Khartoum, Sudan | 16.4% [10.4–28.8] | 16.6% [10.5–29.2] | 0.2% [0.1–1.0] | 53.8% [53.2–56.5] |
| Kigali, Rwanda | 4.5% [2.0–14.3] | 5.2% [2.3–16.8] | 1.6% [0.7–4.4] | 11.2% [10.6–12.1] |
| Kinshasa, Democratic Republic Congo | 17.2% [10.8–27.3] | 17.0% [10.6–26.8] | 22.4% [16.9–48.1] | 46.3% [44.0–49.2] |
| Lagos, Nigeria | 22.9% [15.1–32.6] | 22.0% [15.1–31.7] | 39.5% [29.9–50.4] | 7.7% [6.9–9.0] |
| Luanda, Angola | 43.1% [36.3–51.3] | 43.8% [36.5–52.8] | 41.1% [35.7–45.9] | 36.9% [35.5–39.0] |
| Lubumbashi, Democratic Republic Congo | 51.3% [33.7–67.3] | 53.2% [35.2–69.6] | 41.1% [35.7–45.9] | 46.3% [44.0–49.2] |
| Nakuru, Kenya | 15.1% [11.9–25.7] | 17.9% [15.4–25.2] | 19.0% [8.9–29.1] | 7.1% [6.5–8.1] |
| Ndola, Zambia | - | - | - | 40.1% [37.1–43.7] |
| Oyo, Nigeria | 3.5% [3.2–6.2] | 3.7% [3.5–6.5] | 0.7% [0.4–1.7] | 7.7% [6.9–9.0] |
| Port Elizabeth, South Africa | 47.5% [40.5–52.1] | 63.5% [57.0–68.2] | 39.3% [32.0–43.7] | 5.2% [4.4–6.3] |
| Total | | | | 29.0% [27.1–31.5] |

The national values are taken from ref. 2. The values inside the brackets show the uncertainty due to a possible change in travel speed by 20%.

calculate travel times. All of these products introduce various types of uncertainties into our analysis.

It is acknowledged that gridded population datasets, such as the WorldPop dataset utilized in this analysis, exhibit considerable uncertainty when estimating the number of individuals residing in slums. A comparison of gridded population estimates with demographic health survey data[31] or other data from the literature[32] in different cities in the Global South indicates that the number of individuals residing in informal areas may be ten times higher than suggested by the estimates. In comparing local survey data with gridded population datasets, ref. 38 demonstrated that self-reported population numbers are, on average, up to double the size of gridded population estimates. Consequently, the values calculated for spatial accessibility can be considered a lower boundary for the actual values.

We recognize that the potential incompleteness of OSM data could introduce a degree of bias to our findings. However, OSM street network data have been successfully used to replicate land-use parcels[39] or improve land cover products[40] in several African cities. Moreover, global analyses are quite encouraging and highlight the constant improvement of data quality and completeness[41,42]. Future studies should consider evaluating road completeness, as was recently done by Herfort[43] for building footprints.

The land-use classification we based our analysis on has an accuracy of 71%[23], leaving room for several uncertainties. Nevertheless, the classification of informal areas as such is challenging and a highly debated field. Especially the issue of having adequate ground truth data as well as clearly defining informal settlements from a spatial perspective (What is an informal area, and what is not?) is an open question still under research[7,9,20].

Another limitation comes from the dataset on healthcare facilities. Lists of healthcare facilities in sub-Saharan Africa (SSA) are incomplete[2]. While the data seem reasonable for some of the cities, the number of healthcare facilities in other cities (e.g., Luanda) is considered too low. This clearly affects the results. Moreover, we did not account for localized factors that can affect accessibility, such as informal vendors, lack of maintenance, and

irregular surfaces, as this would require data that is beyond the scope and capacity of our current analysis. However, it is recommended that studies at smaller scales attempt to include these factors to increase the accuracy of the results.

Although information on healthcare facilities is also available from other sources, such as OSM or Google Maps, the dataset we used is the only externally evaluated dataset at a continental scale. One limitation of this study is the exclusive focus on public health facilities within the dataset. This approach does not account for the significant role of private healthcare providers, which, for instance, constitute over 45% of all health facilities in Kenya[44]. It should be noted that the importance of private clinics varies significantly depending on the political and economic context of each country. While this focus on public facilities is a necessary first step in understanding access to healthcare, it may lead to an incomplete picture of the overall healthcare landscape.

Another possible bias in the dataset is that information on healthcare facilities could be outdated (e.g., due to destruction in armed conflicts). Additionally, the capacity of healthcare facilities was omitted from our analysis, limiting our ability to conduct a more nuanced evaluation. Larger facilities with greater capacity typically serve more people, potentially skewing accessibility results if not considered, as they may reduce pressure on smaller, nearby facilities. Conversely, smaller facilities might be overwhelmed, reducing access and increasing wait times, especially in densely populated areas. Incorporating capacity considerations would have enabled a more detailed analysis, such as a two-step floating catchment area approach.

Moreover, the age of the population plays a crucial role, as older populations generally have higher healthcare needs and more frequent service requirements. Areas with older demographics might experience greater strain on local facilities, complicating accessibility further. Future studies should incorporate both facility capacity and population age to enhance the accuracy of healthcare accessibility assessments.

In our analysis, we exclusively examined the aspect of physical accessibility to healthcare facilities. However, it is essential to acknowledge that

healthcare accessibility encompasses various dimensions that extend beyond mere geographic proximity. According to ref. 45, relevant dimensions of access are availability, affordability, and acceptability. Notably, the financial aspect plays a pivotal role, as the costs of treatment may vary significantly among different facilities, potentially influencing individuals to forego the closest option. Another crucial aspect is the social and cultural distance between the healthcare system and its users. Moreover, our analysis lacked granularity in terms of distinguishing between the specialized services offered by healthcare facilities, a crucial factor influencing healthcare-seeking behavior.

By focusing solely on walking, our study provides a conservative estimate of healthcare accessibility, which is particularly relevant in lower-income areas with limited transport options. However, this approach may underestimate actual access for those using other modes of transportation, especially over longer distances where walking is less practical.

This limitation highlights the need for future research to incorporate a broader range of transportation modes, offering a more accurate and comprehensive view of healthcare accessibility in urban areas with diverse transport infrastructures. Despite these limitations, our use of global and continental data products marks an important first step toward systematic inter-urban analyses with meaningful outcomes.

The insights gleaned from our study on SA to healthcare facilities in SSA cities set the stage for promising future research avenues. As we consider the global implications of our findings, extending our analytical framework to a global scale emerges as a pivotal step. Such an expansion would not only facilitate cross-continental comparisons but could also unveil overarching patterns and trends in spatial healthcare accessibility. This broader perspective holds the potential to offer valuable insights for shaping global health planning and policy formulation. To ensure the ongoing relevance and accuracy of our insights, a commitment to integrating the latest available data remains imperative. Regular updates to datasets, encompassing healthcare facility locations, population distribution, and land use information, are essential to capture the dynamic nature of urban landscapes.

As we delve deeper into expanding the scope of our analysis, considering various transportation modes beyond walking could be a major improvement. The inclusion of public transportation networks, road infrastructure, and other relevant transportation modes promises a more nuanced understanding of overall healthcare accessibility in diverse urban settings. Recognizing that healthcare accessibility extends beyond mere physical proximity necessitates the integration of multidimensional travel-time estimates in future studies. This includes considerations of socio-economic accessibility, financial factors, and the availability of specialized healthcare services, contributing to a more comprehensive evaluation of healthcare access.

Integrating information on healthcare facility capacity, including size, capabilities, and services offered, is crucial for refining accessibility analysis. Building on the importance of initiatives like IDEAMaps[9], future research should adopt a holistic approach, analyzing living conditions, community needs, and infrastructural challenges alongside spatial accessibility. Expanding the analysis to include socio-economic factors like income, education, and employment will provide a more comprehensive understanding of disparities and guide equitable healthcare policies. Additionally, using advanced methodologies, such as two-step catchment area analysis, can offer a more nuanced view of healthcare access. By addressing these factors collectively, future studies can provide actionable insights for policymakers and urban planners, enhancing social inclusion and healthcare delivery effectiveness.

## Conclusion
In summary, this study provides important insights into the spatial dynamics of healthcare accessibility in SSA cities, with a specific focus on the disparities between formal and informal urban areas. The findings underscore the critical need for inclusive urban planning that addresses the healthcare needs of all urban residents, particularly those living in informal settlements. As urbanization continues to accelerate in SSA, the challenges highlighted in this study are likely to become more pronounced, making it essential for policymakers to take proactive steps to mitigate these disparities.

Healthcare accessibility is a key determinant of public health, and disparities in access can have significant consequences for health equity. The study also emphasizes the importance of using spatial analysis as a tool for identifying and addressing inequalities in urban health provisioning. By highlighting the gaps in healthcare access within SSA cities, this research provides a foundation for further studies and policy interventions aimed at achieving more equitable health outcomes across rapidly growing urban areas.

## Data availability
All the data used to perform the analysis can be openly accessed. The gridded population data from WorldPop is available at https://hub.worldpop.org/geodata/listing?id=79. The data on healthcare facilities is described in the following publication: https://www.nature.com/articles/s41597-019-0142-2. The data on land use in the analysed cities is available at https://code.earthengine.google.com/?asset=projects/wri-datalab/urban_land_use/V1. The OSM data can easily be accessed via different API's (e.g., in QGis). The CSV-files containing the merged information on the service areas are stored in a Zenodo Repository[46].

## Code availability
The code to perform the analyses, to create the figures and the tables (e.g., calculate the relative risks) is available in a Zenodo Repository[46].

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

## Acknowledgements

This document is a result of the research project "Uniform detection and modeling of slums to determine infrastructure needs" funded by the LOEWE Program of Hesse State Ministry for Higher Education, Research and the Arts. This work has been co-funded by the LOEWE initiative (Hesse, Germany) within the emergenCITY center [LOEWE/1/12/519/03/05.001(0016)/72]. S.G. and J.H. were supported by the Formas grant [2023-01210], specifically during the revision and finalization stages of the manuscript.

## Author contributions

J.F. conducted the analysis, prepared the figures, and wrote the original draft. S.G. merged service areas with population data and reviewed and edited the paper. J.H. calculated the service areas and reviewed and edited the paper.

## Funding

## Competing interests

The authors declare no competing interests.
