## [Transparent Peer Review File · Communications Medicine]

Differences in walking access to healthcare facilities between formal and informal areas in 19 sub-Saharan African cities

Corresponding Author: Dr John Friesen

Version 0:

Reviewer comments:

Reviewer #1

(Remarks to the Author)

General Comments

The manuscript, "Differences in walking access to health-care services between formal and informal areas in 19 sub-Saharan cities", investigated spatial access in the form of walking to healthcare facilities in selected sub-Saharan African cities. The rationale of this manuscript is quite a relevant, as it provides a consistent framework to examine the contextual drivers that facilitate or hinder walking access to healthcare facilities. The writing is relatively well-structured and organized. There are some grammatical and typographical errors that need to be corrected. Also, the presentation needs to be improved at some sections. The following specific issues need to be addressed.

Specific comments

- a). Consider revising the title by changing the phrase, 'sub-Saharan cities' to 'sub-Saharan African cities'.
- b). Table 1 and line 222, page 5: correct the spelling for 'Addis Abeba'. The correct spelling is 'Addis Ababa'. Correct this in the entire document.
- c). Line 312: Correct the grammar in this sentence "Values significantly higher than 1 indicate that...". There are similar grammatical errors in the document, please correct accordingly.
- d). Line 447: There are several cases of sentences beginning with numbers. Please, revise this in the entire document.
- e). The authors have rightly acknowledged the study limitations and identified areas for future research. However, there is existing data on economic status data based on Relative Wealth Index (RWI) estimated by Chi et al. (2022) at 2.4km spatial resolution that can be obtained from the Humanitarian Data Exchange. This has been used in the study by Dumedah et al. (2023) in relation to spatial access to healthcare facilities, similar to this study.
Gift Dumedah, Seidu Iddrisu, Christabel Asare, Samuel Adu-Prah, Sinead English, Inequities in spatial access to health services in Ghanaian cities, Health Policy and Planning, Volume 38, Issue 10, December 2023, Pages 1166–1180, <https://doi.org/10.1093/heapol/czad084>.
Chi G, Fang H, Chatterjee S, Blumenstock JE. 2022. Microestimates of wealth for all low- and middle-income countries. Proceedings of the National Academy of Sciences 119: e2113658119.
- f). Formal and informal urban settlements/communities. This is a key concept in this study, yet the authors have not adequately addressed its conceptual framing and its specific aspect that was used in this study. This study seemed to adopt the urban morphological description, but it is important to acknowledge other descriptions and how this urban form depiction may actually encompass other alternatives drivers.
- g). The same applies to spatial access as well, framing of this concept, the specific measurement approach under which direct walking access applies, and acknowledge of alternative measurement approaches.

Reviewer #2

(Remarks to the Author)

This is a high quality paper on an important and highly relevant topic. I have a few minor critiques and suggestions.

Has there been any on the ground validation of WorldPop data with regard to 'informal settlements'? These data are relatively good for large areas on average - but may not be that accurate at more granular scales, and it would be good to discuss any work that has looked at how well the WorldPop data correspond to the types of settlements described here.

Does the SSA health facility dataset include private clinics? I believe it does not. Depending on location, this could mean that many private clinics exist in reality but not in this analysis.

Line 119 - why was a 15km buffer used around the cities?

How were these 19 cities chosen?

Line 149 - please define each of these categories. "Atomistic" won't make sense for many readers.

Line 167 - Suggest indicating "F" and "IF" for formal and informal, if that is what is being referred to in the formula for RR on line 169

The relative risk measure is nice and creative. It would of course be sensitive to the LULC data and the WorldPop data, but the measures for the entire city are probably less so.

The discussion section really could use more discussion. Why is this topic relevant?

I suggest some concluding remarks other than the researchers' next steps. What should the reader take away from this in summary?

Reviewer #3

(Remarks to the Author)

This manuscript titled "Differences in walking access to health-care services between formal and informal areas in 19 sub-Saharan cities" offers a very much-needed large-scale accessibility analysis to healthcare in sub-Saharan cities. The contributions proposed are highly relevant to development goals and could establish a benchmark for future studies and policy documents. However, there are several aspects that I recommend clarifying before publication. These would confirm/increase the credibility of the findings. Please see my comments below.

General comments:

- It is surprising to see the limited discussion around the foundational aspects of accessibility and accessibility to healthcare, see crucially: (Levinson & Wu, 2020; Lucas et al., 2019; Martens et al., 2019). This should also be accompanied by an operational definition.
- Similarly, the discussion about the implications of access to healthcare facilities should be strengthened (e.g. why this matters?). Some pieces can help to illustrate the point, e.g. (Cabrera-Barona et al., 2018; Chen et al., 2023).
- The selected sample of 19 cities should be in perspective at least the continental or subcontinental level, e.g. what the representation, or influence of this sample?
- Developing regions often experience several major physical obstacles in the infrastructure, such as informal vendors, lack of maintenance, and irregular surfaces e.g. (Trichês Lucchesi et al., 2023). How is this considered in the analysis? Additionally, there are perceived issues that are especially prevalent in this context, e.g., crime (Evans, 2009).
- I found a few typos, e.g. line 312, "higher [than] 1". I recommend double-checking the text.

Introduction

- As mentioned before, there is a lack of discussion on the definition of accessibility and why it matters.
- An additional point in favour of urban areas in Africa is the expected population growth from a global perspective. This can be highlighted here.
- A further example of accessibility to health services can be found in (Verduzco Torres & McArthur, 2024).

Methods

- A point to consider walking only (and not other modes) might be that this represents a lower-bound accessibility level. The authors might consider emphasizing this.
- A 180-minute threshold is a lot. Are there any references on observed travel times?
- From the manuscript, it is not clear accessibility to whom? The standard 5 kph speed suggests it is a healthy population. What are the implications of this in the findings? In any case, further discussion is needed based on the evidence in the literature.
- Topography is demonstrated to be an important factor affecting walking time. How does this vary from city to city? How is this considered in the analysis? If it is not, measures of uncertainty would be needed.

- What are the parameters and rules used for walking routes? In developing context, the limited pedestrian infrastructure can make residents to rely on road infrastructure. How is this considered in the routing model?
- It would be important to briefly discuss the key aspects of the healthcare data in more detail. This is a crucial input of the analysis and would increase the credibility and robustness, e.g. collection methods, validation, and coverage, time period.
- Additionally, the authors mention that they have “categorized them into two groups according to the table provided as supplementary material. While the first group (i) refers to hospitals and centres with surgical care, the second group (ii) includes health clinics and other small health facilities.” This should be elaborated further, what is the key criteria employed and what is the rationale behind?
- How does the size/capacity of the facility might influence the results? This is partially acknowledged in the limitations. Can the authors elaborate how is this expected to affect the results? Age of the population can play a role.
- What are the considerations taken to assess/trust the completeness of the OSM road network data? This might bias the findings substantially. Are there references for the cases studied? This has been done for building data in (Herfort et al., 2023).
- Could the authors clarify the rationale and interpretation of the relative risk in lines 168 to 169?

Results

- Section 3.1 could re-iterate the type of health-care facilities included in to improve clarity for readers.
- Figure 1, can the authors clarify the interpretation of error bars?
- Figure 2 is hard to read; could it use annotation? Also, removing error shade can improve clarity. Also, the authors are encouraged to consider using an accessible blind-colour palette.
- The header of section 3.2 should reflect that this is about residential land use only (locations of hospitals is land use too).
- Figure 3, adding column and rows labels would help to increase clarity.
- In figure 3, top-right panel, there is a big gap between the three cities with lowest accessibility and all of the rest. Is there an explanation from authors?
- Table 2 and 3, including the meaning of intervals in square brackets in a footnote or caption can improve readability.

Discussion

- The authors are claiming to confirm the importance of medium-size cities. But it is not very clear how these type of cities play a relevant role based on the results. Could the authors clarify?

References

- Cabrera-Barona, P., Blaschke, T., & Gaona, G. (2018). Deprivation, Healthcare Accessibility and Satisfaction: Geographical Context and Scale Implications. *Applied Spatial Analysis and Policy*, 11(2), Article 2. <https://doi.org/10.1007/s12061-017-9221-y>
- Chen, H., Cao, Y., Feng, L., Zhao, Q., & Verduzco Torres, J. R. (2023). Understanding the spatial heterogeneity of COVID-19 vaccination uptake in England. *BMC Public Health*, 23(1), 895. <https://doi.org/10.1186/s12889-023-15801-w>
- Evans, G. (2009). Accessibility, Urban Design and the Whole Journey Environment. *Built Environment*, 35(3), 366–385. <https://doi.org/10.2148/benv.35.3.366>
- Herfort, B., Lautenbach, S., Porto de Albuquerque, J., Anderson, J., & Zipf, A. (2023). A spatio-temporal analysis investigating completeness and inequalities of global urban building data in OpenStreetMap. *Nature Communications*, 14(1), 3985. <https://doi.org/10.1038/s41467-023-39698-6>
- Levinson, D. M., & Wu, H. (2020). Towards a general theory of access. *Journal of Transport and Land Use*, 13(1), Article 1. <https://doi.org/10.5198/jtlu.2020.1660>
- Lucas, K., Martens, K., Di Ciommo, F., & Dupont-Kieffer, A. (2019). Introduction. In K. Lucas, K. Martens, F. Di Ciommo, & A. Dupont-Kieffer (Eds.), *Measuring Transport Equity* (pp. 3–12). Elsevier. <https://doi.org/10.1016/B978-0-12-814818-1.00001-9>
- Martens, K., Bastiaanssen, J., & Lucas, K. (2019). Measuring transport equity: Key components, framings and metrics. In *Measuring Transport Equity* (pp. 13–36). Elsevier. <https://doi.org/10.1016/B978-0-12-814818-1.00002-0>
- Trichês Lucchesi, S., de Abreu e Silva, J., Margarita Larranaga, A., Zechin, D., & Beatriz Bettella Cybis, H. (2023). Machine Learning and Image Recognition Technologies to Identify Built Environment Barriers and Incentives to Walk. *Transportation Research Record*, 2677(1), 14–24. <https://doi.org/10.1177/03611981221097965>
- Verduzco Torres, J. R., & McArthur, D. P. (2024). Public transport accessibility indicators to urban and regional services in Great Britain. *Scientific Data*, 11(1), Article 1. <https://doi.org/10.1038/s41597-023-02890-w>

Reviewer #4

(Remarks to the Author)

The authors present an interesting approach to a very complicated issue. We have known for a long time that patterns of spatial accessibility in urban areas are very different from those in rural areas, but addressing this problem in research remains a challenge to this day because many traditional approaches, such as the raster-based approach, are unfit for urban areas. The authors have used a methodology to assess this complicated issue. The paper is interesting to read and addresses an important issue. The authors have thought about many of the assumptions that feed into this methodology, such as “how do you define a city?” and “how do you separate informal areas and formal areas?”. However, in my opinion, some important challenges still need to be addressed to make the paper more robust and publishable.

Major concerns:

1) The authors have used the network analyst approach. This approach needs a continuous/connected road network without interruptions or missing segments. In my opinion, this is especially hard to achieve in informal settlements where OpenStreetMap hasn't been able to digitize all road segments yet. The authors have not mentioned how they ensured that all road segments are connected and what they did to minimize bias in missing road segments in general and particularly in informal settlements? Have they considered the AI digitized road segments from data for Good from Meta? (<https://github.com/facebookmicrosites/Open-Mapping-At-Facebook/wiki/Available-Countries>)

2) The authors have chosen a walking speed of 5km/h which is quite high in the case of people needing hospital care. How did the authors come up with this speed and on which target population was it based? I would suggest the authors to lower this speed and also have a look at the following paper that discusses walking speeds for mothers and children in more depth: Watmough, G.R., Hagdorn, M., Brumhead, J. et al. Using open-source data to construct 20 metre resolution maps of children's travel time to the nearest health facility. *Sci Data* 9, 217 (2022). <https://doi.org/10.1038/s41597-022-01274-w>.

3) The authors make quite large assumptions on the classification of specific types of health facilities into two groups and say that one of the groups has surgical capacity. Unfortunately, many of these assumptions have proven to be impossible because types of health facilities are rarely a good indication of the types of services they provide, see (Petragallo S, Timoner P, Hierink F, et al. Assessing the accuracy of health facility typology in representing the availability of health services: a case study in Mali. *BMJ Open* 2024;14:e077127. doi: 10.1136/bmjopen-2023-077127). What is the main purpose of dividing these facilities into these groups? I think that the overall statistics are already interesting in itself without the sub-classification.

4) In several places in the manuscript the authors compare their results to previous studies, such as Weiss et al. and Ouma et al. However, the raster-based approach in these studies also includes rural areas and the approach also allows speeds for other types of landcovers. In addition, at the time of these studies only the unconstrained data was available for the gridded population, meaning that a comparison in the statistics is complicated. I would highly recommend the authors to look at papers that assessed access to healthcare specifically in urban areas and using different assessment techniques, such as the work recently published by Banke-Thomas, Aduragbemi, et al. "Geographical accessibility to functional emergency obstetric care facilities in urban Nigeria using closer-to-reality travel time estimates: a population-based spatial analysis." *The Lancet Global Health* 12.5 (2024): e848-e858. How does their approach and findings relate to these results, are their coverage stats higher/lower and why is that?

5) The authors used WorldPop constrained UN adjusted data. I would recommend them to mention the uncertainty of estimating population in informal settlements in the limitation section, as mentioned in these two articles:

• Thomson, D.R.; Gaughan, A.E.; Stevens, F.R.; Yetman, G.; Elias, P.; Chen, R. Evaluating the Accuracy of Gridded Population Estimates in Slums: A Case Study in Nigeria and Kenya. *Urban Sci.* 2021, 5, 48.

<https://doi.org/10.3390/urbansci5020048>

• Breuer, Julius HP, et al. "The unseen population: Do we underestimate slum dwellers in cities of the Global South?." *Habitat International* 148 (2024): 103056.

Minor concerns:

1) Line 57: "SA is addressed as an indicator in the sustainable development goal (SDG) 3.8." it is implied but not specifically mentioned nor formulated as an indicator or measurement. Suggestion to reformulate.

2) Line 80-81, there are increasing publications on urban access to care in SSA. See articles:

a. Macharia, Peter M., et al. "A geospatial database of close-to-reality travel times to obstetric emergency care in 15 Nigerian conurbations." *Scientific data* 10.1 (2023): 736.

b. Geographical accessibility to functional emergency obstetric care facilities in urban Nigeria using closer-to-reality travel time estimates: a population-based spatial analysis." *The Lancet Global Health* 12.5 (2024): e848-e858.

3) Line 108-109: "How good is the walking accessibility to health-care facilities for different cities in SSA and how does this compare to global or national estimates?" I would recommend the authors to reduce this comparison to other comparable studies that have looked at urban contexts or used a similar method as indicated in the major concerns.

4) Line 169: RR = SAF/SAIF, can the others at the abbreviation for "F" and "IF" above? Such as Formal (F) and Informal (IF)?

5) Table 1, last column. If only counting the number of hospitals, I would suggest renaming the column to number of hospitals instead of health facilities (just hospitals).

6) Line 258-261: You mean the second closest within 15 minutes of the first closest? Please rephrase.

7) Caption for figure two: I think it would be easier to mention the health facilities in group 2 e.g. clinics and smaller health facilities + mention whether it is in full urban areas or just in formal or informal settlements.

8) Line 312: you miss the word "than", significantly higher "than" 1.

Version 1:

Reviewer comments:

Reviewer #1

(Remarks to the Author)

The authors have adequately addressed all my comments. Also, they have improved the quality of the manuscript through this revision. I believe the current manuscript is acceptable for publication.

Reviewer #2

(Remarks to the Author)

The authors have largely addressed all of my comments. I have two minor further comments that I believe would help increase clarity for their results.

The authors added some text to justify the 15km buffer around cities, but didn't specify why 15km was chosen. Why not 16km or 14km? Just a simple statement would suffice. I believe it is OK if it is relatively arbitrary, but was chosen to reach outside of city limits.

The authors also added some text to explain why the 19 cities were chosen (they were the 19 cities within SS Africa from the previous analysis). But, I think it would still be useful to specifically say this. Perhaps something along the lines of: These 19 villages were all villages from the previous analysis (reference) that were located in SS Africa.

Reviewer #3

(Remarks to the Author)

I would like to thank the authors for addressing the comments and questions raised in the previous round of comments thoroughly. They have made serious considerations to improve the manuscript.

My only minor comment with the updated version is the following:

Please use annotations in Figure 2. I understand it might be difficult to annotate all curves. However, they can focus on a few cases that need to be highlighted to readers at least. Otherwise, it is very difficult to read the figure and therefore does not contribute to the paper. Also, could the authors confirm if they are using an accessible colour-blind palette? If not, why?

Reviewer #4

(Remarks to the Author)

Thank you for the reviews which have notably improved the paper. The authors have sufficiently addressed all my comments.

Reviewer #1 (Remarks to the Author):

General Comments

The manuscript, “ Differences in walking access to health-care services between formal and informal areas in 19 sub-Saharan cities”, investigated spatial access in the form of walking to healthcare facilities in selected sub-Saharan African cities. The rationale of this manuscript is quite a relevant, as it provides a consistent framework to examine the contextual drivers that facilitate or hinder walking access to healthcare facilities. The writing is relatively well-structured and organized. There are some grammatical and typographical errors that need to be corrected. Also, the presentation needs to be improved at some sections. The following specific issues need to be addressed.

Specific comments

a). Consider revising the title by changing the phrase, ‘sub-Saharan cities’ to ‘sub-Saharan African cities.

Thank you for this comment. We changed the title according to your suggestion.

b). Table 1 and line 222, page 5: correct the spelling for ‘Addis Abeba’. The correct spelling is ‘Addis Ababa’. Correct this in the entire document.

We changed the spelling in the text.

c). Line 312: Correct the grammar in this sentence “Values significantly higher than 1 indicate that...”. There are similar grammatical errors in the document, please correct accordingly.

Thank you for the comment. We changed the sentence and carefully revised the manuscript.

d). Line 447: There are several cases of sentences beginning with numbers. Please, revise this in the entire document.

Thank you for this comment. We updated the document according to your comments.

e). The authors have rightly acknowledged the study limitations and identified areas for future research. However, there is existing data on economic status data based on Relative Wealth Index (RWI) estimated by Chi et al. (2022) at 2.4km spatial resolution that can be obtained from the Humanitarian Data Exchange. This has been used in the study by Dumedah et al. (2023) in relation to spatial access to healthcare facilities, similar to this study.

Gift Dumedah, Seidu Iddrisu, Christabel Asare, Samuel Adu-Prah, Sinead English, Inequities in spatial access to health services in Ghanaian cities, Health Policy and Planning, Volume 38, Issue 10, December 2023, Pages 1166–1180, <https://doi.org/10.1093/heapol/czad084>.

Chi G, Fang H, Chatterjee S, Blumenstock JE. 2022. Microestimates of wealth for all low- and middle-income countries. Proceedings of the National Academy of Sciences 119: e2113658119.

Thank you for this very important point. We incorporated the study you mentioned and another study to highlight the already existing approach in this regard:

“Banke-Thomas et al. \cite{banke2024geographical} took a first step towards intra-urban analysis of access to maternity care in SSA by analysing travel times for 15 Nigerian cities. They found that informal settlements appear to be worse off compared to parts of the inner city.

Dumedah et al. \cite{dumedah2023inequities} analysed the access to healthcare in 4 Ghanaian cities. Similar to the studies mentioned above, they find that relative wealth negatively correlates with access to healthcare in the cities investigated.”

f). Formal and informal urban settlements/communities. This is a key concept in this study, yet the authors have not adequately addressed its conceptual framing and its specific aspect that was used in this study. This study seemed to adopt the urban morphological description, but it is important to acknowledge other descriptions and how this urban form depiction may actually encompass other alternatives drivers.

Thank you for bringing up this important point. We were unclear in this regard and have improved our wording in many cases. We have added significant additional information on the fact that we used urban morphology as a proxy for poverty. There are of course many other definitions possible dealing with this concept.

“As mentioned previously, we rely on urban morphology as a proxy for the social groups living in the respective parts of the city. Urban morphology, which refers to the physical form and structure of urban environments, is a well-established approach in urban studies to distinguish between different types of settlements. This approach is particularly useful when direct socio-economic data are unavailable, difficult to obtain, inconsistent throughout different countries or have lower spatial resolutions (e.g. data from demographic health surveys). In this study, urban morphology is leveraged to differentiate between formal and informal urban settlements, as the physical characteristics of these areas—such as building density, layout, and infrastructure—often reflect underlying socio-economic conditions.”

“The distinction between formal and informal settlements is critical to understanding urban inequalities. Formal settlements are generally planned and regulated, with better access to services and infrastructure, whereas informal settlements often emerge spontaneously and lack such resources. These differences in urban form are not just physical; they are deeply intertwined with socio-economic factors such as income levels, employment types, and access to public services. Nonetheless, we justify the use of urban morphology in this study due to the lack of comprehensive socio-economic data at the necessary spatial scale across the cities we examined. Guzder Williams et al. \cite{GuzderWilliams.2023} classification provides a standardized, globally applicable framework that allows for consistent comparison across diverse urban contexts. This approach is particularly useful in rapidly urbanizing regions where data collection can be challenging, and where urban form often remains a strong indicator of different social groups \cite{Wurm.2018}.”

g). The same applies to spatial access as well, framing of this concept, the specific measurement approach under which direct walking access applies, and acknowledge of alternative measurement approaches.

Thank you for this important aspect, which was also brought up from another reviewer. We framed our approach to measure access in more general way by including the following paragraphs into our methods section:

“Access to a place or service can be assessed along a number of dimensions, as outlined in a comprehensive theory by Levinson and Wu \cite{levinson2020towards}. It starts with (i) where to measure access, (ii) what kind of barrier or cost to consider, (iii) how to assess access, (iv) what time of day to assess access, (v) what place to assess access, (vi) what mode of transport to consider, and (vii) what group of people to consider \cite{levinson2020towards}.”

We applied aspects of Levinson and Wu’s theory to define healthcare access in our study, focusing on (i) urban areas in SSA, (ii) travel time via (iii) isochrones. Due to data scarcity, we (iv) did not account for daily fluctuations. The study focused on (v) healthcare facilities, using (vi) walking as the travel mode, and (vii) comparing urban groups based on urban morphology, a proxy for living conditions shown to correlate with social groups \cite{Wurm.2018}.”

We also acknowledged other measurement approaches, like friction surfaces and explained our choice in more detail:

“We opted to use street networks instead of friction surfaces (like Ouma et al. \cite{Ouma.2018}) to calculate travel time because they provide a more accurate reflection of pedestrian movement within urban environments. Street networks capture the actual routes people take, considering the layout and connectivity of roads, which is crucial in dense urban areas, especially where informal settlements are prevalent. While friction surfaces offer a broad approximation, they often lack the precision needed to represent the complexities of urban movement. Using street networks ensures a more realistic assessment of healthcare accessibility in SSA.”

Additionally, we also highlighted this aspect in our discussion section:

“By focusing solely on walking, our study provides a conservative estimate of healthcare accessibility, which is particularly relevant in lower-income areas with limited transport options. However, this approach may underestimate actual access for those using other modes of transportation, especially over longer distances where walking is less practical.

This limitation highlights the need for future research to incorporate a broader range of transportation modes, offering a more accurate and comprehensive view of healthcare accessibility in urban areas with diverse transport infrastructures. Despite these limitations, our use of global and continental data products marks an important first step toward systematic inter-urban analyses with meaningful outcomes.”

Reviewer #2 (Remarks to the Author):

This is a high quality paper on an important and highly relevant topic. I have a few minor critiques and suggestions.

Has there been any on the ground validation of WorldPop data with regard to 'informal settlements'? These data are relatively good for large areas on average - but may not be that accurate at more granular scales, and it would be good to discuss any work that has looked at how well the WorldPop data correspond to the types of settlements described here.

Thank you for this comment. We are aware of the uncertainties of gridded population data sets, especially in the context of informal settlements and added two extra paragraphs (one in the methods section and one in the limitations section) to emphasize the importance of this aspect.

“Although the data sets used here can be highly uncertain, especially in the area of informal settlements (cf. \cite{thomson2021evaluating, breuer2024unseen}), they are nevertheless the best data sets available on a continental scale and thus make comparative studies possible in the first place.”

“It is acknowledged that gridded population data sets, such as the WorldPop data set that was utilised in the analysis, exhibit considerable uncertainty when estimating the number of individuals residing in slums. A comparison of gridded population estimates with demographic health survey data \cite{thomson2021evaluating} or other data from the literature \cite{breuer2024unseen} in different cities in the Global South indicates that the number of individuals residing in informal areas may be ten times higher than suggested by the estimates. In comparing local survey data with gridded population data sets, Abascal et al. \cite{abascal2024making} demonstrated that self-reported population numbers are, on average, up to double the size of gridded population estimates. Consequently, the values calculated for spatial accessibility can be considered a lower boundary for the actual values.”

Does the SSA health facility dataset include private clinics? I believe it does not. Depending on location, this could mean that many private clinics exist in reality but not in this analysis.

Thank you for your insightful comment regarding the inclusion of private clinics in the SSA health facility dataset. We agree that the omission of private healthcare facilities could potentially limit the comprehensiveness of our analysis, particularly in regions where private providers play a significant role in healthcare delivery.

We acknowledge that in countries like Kenya, private healthcare providers account for a substantial portion of health services, and this dynamic can vary widely across different political and economic contexts within sub-Saharan Africa. To address this, we highlighted in our manuscript the limitation of focusing solely on public health facilities, and how this approach, while a necessary first step, might not capture the full spectrum of healthcare accessibility.

“Although information on health-care is also available from other sources such as OSM or Google Maps, the data set we used is the only externally evaluated data set at continental scale. One limitation of this study is the exclusive focus on public health facilities within the dataset. This approach does not account for the significant role of private healthcare providers, which, for instance, constitute over 45% of all health facilities in Kenya \cite{barnes2010private}. Here it has to be noticed that the importance of private clinics varies significantly depending on the political and economic context of each country. While this focus on public facilities is a necessary first step in understanding access to healthcare, it may lead to an incomplete picture of the overall healthcare landscape.”

Besides, we are planning a follow-up study that will include data from local sources in Kigali, Rwanda, where we aim to integrate both public and private healthcare facilities. This study will provide a more detailed and context-specific analysis, allowing us to better understand the role of private clinics in urban healthcare accessibility.

Line 119 - why was a 15km buffer used around the cities?

We used a 15 km buffer to account for health facilities lying outside of the city limits but can be reached faster than facilities within the city boundary.

We added the following paragraph to make our point more clear:

“The network was built using OSM data, including a 15-km buffer around city boundaries to account for health facilities that are not within city boundaries but can be reached more quickly than facilities within the city.”

How were these 19 cities chosen?

Thank you for this comment. In the original paper of Guzder-Williams et al. (who performed the land use classification), 200 cities worldwide were presented. We used information on all 19 cities in sub Saharan Africa available in this data set and excluded no city. We updated the text in the methods section in the following way:

“As mentioned previously, to distinguish between different types of land use within city boundaries, we used the classification of Guzder Williams et al. \cite{GuzderWilliams.2023}. Their classification is based on a machine learning algorithm and has a resolution of around 5 m and classified more than 200 cities worldwide. The group aims for a classification of all cities worldwide with more than 100 000 inhabitants and the published dataset is a first subset with exemplary cities in different countries and with various sizes.”

Line 149 - please define each of these categories. "Atomistic" won't make sense for many readers.

Thank you for this comment. We added an additional paragraph and explained the different land uses in more detail, referring to their definition in the original paper:

“Guzder Williams et al. \cite{GuzderWilliams.2023} give the following definitions of the different categories: Atomistic settlements are "areas occupied for settlement before planning had occurred". Informal land subdivision refer to "areas occupied for settlement and presumed planned informally, based on visible infrastructure quality, parcel sizes, road widths, and connection to arterial roads". Formal land subdivisions are "areas occupied for settlement and presumed planned formally, with approval of the municipal government, based on visible infrastructure quality, parcel sizes, road widths, and connections to arterial roads". Finally, housing projects are "areas occupied for settlement with home construction and land subdivision conducted under the same plan, based on layout and similarity of structures" \cite{GuzderWilliams.2023}.”

Line 167 - Suggest indicating "F" and "IF" for formal and informal, if that is what is being referred to in the formulat for RR on line 169

Thank you for this comment. We added information to explain the meaning of the indices.

The relative risk measure is nice and creative. It would of course be sensitive to the LULC data and the WorldPop data, but the measures for the entire city are probably less so.

Thank you for this remark. We agree on your view.

The discussion section really could use more discussion. Why is this topic relevant?

Thank you for this important und helpful comment. We added three small paragraphs into our discussion to highlight the importance of the study and its value:

“The relevance of this study extends beyond the immediate findings on spatial accessibility (SA) to healthcare facilities in Sub-Saharan African (SSA) cities. The disparities highlighted between formal and informal urban settlements underscore broader issues of inequality that are crucial to urban planning and public health. In rapidly urbanizing regions, where informal settlements continue to expand, ensuring equitable access to essential services like healthcare is a pressing challenge. These

findings underscore the need for urban planners and policymakers to prioritize the inclusion of informal areas in urban development plans, particularly in healthcare provisioning.

Urban planning often prioritizes formal areas due to their regulated nature and easier integration into city-wide infrastructure networks. However, as this study demonstrates, neglecting informal settlements can exacerbate existing health inequities, leaving large segments of the urban population vulnerable to disease outbreaks and other health crises. The study's findings serve as a critical reminder that achieving health equity in urban settings requires an inclusive approach that addresses the needs of all urban residents, regardless of the formal status of their living conditions.

The disparities in healthcare accessibility revealed by this study have profound implications for public health outcomes in SSA cities. Limited access to healthcare facilities in informal settlements not only increases the risk of untreated illnesses but also contributes to broader public health challenges, such as the spread of infectious diseases. The relative risk analysis indicating that residents of informal settlements are more likely to experience limited access to healthcare reinforces the urgent need for targeted public health interventions in these areas.”

I suggest some concluding remarks other than the researchers' next steps. What should the reader take away from this in summary?

Thank you for this valuable comment. Beside the additional discussion, we also added a short conclusion, to not end the paper with possible future work, but to highlight the findings, the reader should take away:

“In summary, this study provides important insights into the spatial dynamics of healthcare accessibility in SSA cities, with a specific focus on the disparities between formal and informal urban areas. The findings underscore the critical need for inclusive urban planning that addresses the healthcare needs of all urban residents, particularly those living in informal settlements. As urbanization continues to accelerate in SSA, the challenges highlighted in this study are likely to become more pronounced, making it essential for policymakers to take proactive steps to mitigate these disparities.

Healthcare accessibility is a key determinant of public health, and disparities in access can have significant consequences for health equity. The study also emphasizes the importance of using spatial analysis as a tool for identifying and addressing inequalities in urban health provisioning. By highlighting the gaps in healthcare access within SSA cities, this research provides a foundation for further studies and policy interventions aimed at achieving more equitable health outcomes across rapidly growing urban areas.”

Reviewer #3 (Remarks to the Author):

This manuscript titled “Differences in walking access to health-care services between formal and informal areas in 19 sub-Saharan cities” offers a very much-needed large-scale accessibility analysis to healthcare in sub-Saharan cities. The contributions proposed are highly relevant to development goals and could establish a benchmark for future studies and policy documents. However, there are several aspects that I recommend clarifying before publication. These would confirm/increase the credibility of the findings. Please see my comments below.

General comments:

- It is surprising to see the limited discussion around the foundational aspects of accessibility and accessibility to healthcare, see crucially: (Levinson & Wu, 2020; Lucas et al., 2019; Martens et al., 2019). This should also be accompanied by an operational definition.

Thank you for this valuable and important comment. Based on the holistic approach of Levinson and Wu we developed an operational definition of access to health care. We also discussed various dimensions of access to healthcare in the limitations section:

“Access to a place or service can be assessed along a number of dimensions, as outlined in a comprehensive theory by Levinson and Wu \cite{levinson2020towards}. It starts with (i) where to measure access, (ii) what kind of barrier or cost to consider, (iii) how to assess access, (iv) what time of day to assess access, (v) what place to assess access, (vi) what mode of transport to consider, and (vii) what group of people to consider \cite{levinson2020towards}.

Out of this general theory we take several aspects to build an operational definition of access to healthcare our study is based on. We performed our analysis (i) urban areas in SSA, focusing on (ii) traveltime by calculating (iii) isochrones. Due to data scarcity we (iv) didn't consider daily fluctuations in availability or travel impedance (e.g. traffic during the mornings). We focused on (v) healthcare facilities, used (vi) walking as mode of travel and (vii) compared different urban groups, based on the morphology of the urban area, they live in.”

“In our comprehensive spatial accessibility analysis focused on health-care, we exclusively examined the aspect of physical accessibility to health-care facilities.

However, it is essential to acknowledge that health-care accessibility encompasses various dimensions that extend beyond mere geographic proximity. According to Gilson et al. \cite{gilson2007challenging}, relevant dimensions of access are availability, affordability and acceptability.

Notably, the financial aspect plays a pivotal role, as the costs of treatment may vary significantly among different facilities, potentially influencing individuals to forego the closest option. Another crucial aspect is the social and cultural distance between the healthcare system and their users. Moreover, our analysis lacked granularity in terms of distinguishing between the specialized services offered by health-care facilities, a crucial factor influencing health-care-seeking behavior.”

- Similarly, the discussion about the implications of access to healthcare facilities should be strengthened (e.g. why this matters?). Some pieces can help to illustrate the point, e.g. (Cabrera-Barona et al., 2018; Chen et al., 2023).

Thank you very much for this important comment. We strengthen the implications of our findings in some additional paragraphs within our discussion and also used references you mentioned.

“Urban planning often prioritizes formal areas due to their regulated nature and easier integration into city-wide infrastructure networks. However, as this study demonstrates, neglecting informal settlements can exacerbate existing health inequities, leaving large segments of the urban population vulnerable to disease outbreaks and other health crises. The study's findings serve as a critical

reminder that achieving health equity in urban settings requires an inclusive approach that addresses the needs of all urban residents, regardless of the formal status of their living conditions \cite{cabrera2018deprivation}.”

- The selected sample of 19 cities should be in perspective at least the continental or subcontinental level, e.g. what the representation, or influence of this sample?

Thank you for this comment. In the original paper of Guzder-Williams et al. (who performed the land use classification), 200 cities worldwide were presented. We used information on all 19 cities in sub Saharan Africa available in this data set and excluded no city. We updated the text in the methods section in the following way:

“As mentioned previously, to distinguish between different types of land use within city boundaries, we used the classification of Guzder Williams et al. \cite{GuzderWilliams.2023}. Their classification is based on a machine learning algorithm and has a resolution of around 5 m and classified more than 200 cities worldwide. The group aims for a classification of all cities worldwide with more than 100 000 inhabitants and the published dataset is a first subset with exemplary cities in different countries and with various sizes.”

- Developing regions often experience several major physical obstacles in the infrastructure, such as informal vendors, lack of maintenance, and irregular surfaces e.g. (Trichês Lucchesi et al., 2023). How is this considered in the analysis? Additionally, there are perceived issues that are especially prevalent in this context, e.g., crime (Evans, 2009).

Thank you for raising this point. While these are important considerations (Trichês Lucchesi et al., 2023; Evans, 2009), our study focuses on a broader, city-wide scale to assess general patterns of spatial accessibility across multiple cities. Including these detailed, localized factors would require data that is beyond the scope and capacity of our current analysis.

Our approach aims to maintain comparability across different urban contexts, and while we recognize the value of addressing these micro-level issues, they would be best suited for more localized studies. Future research could explore these factors in greater detail. We include this point in the discussion section of the manuscript.

„Moreover, we did not account for localized factors that can affect accessibility such as informal vendors, lack of maintenance, and irregular surfaces, as this would require data that is beyond the scope and capacity of our current analysis. However, it is recommended that studies at smaller scales attempt to include this factors to increase the veracity of the results.“

- I found a few typos, e.g. line 312, “higher [than] 1”. I recommend double-checking the text. Thank you for your comment. We carefully revised the manuscript.

Introduction

- As mentioned before, there is a lack of discussion on the definition of accessibility and why it matters.

Please see our comprehensive answer above.

- An additional point in favour of urban areas in Africa is the expected population growth from a global perspective. This can be highlighted here.

Thank you for this comment. We added a little more information on this topic in the introduction and in the discussion.

- A further example of accessibility to health services can be found in (Verduzco Torres & McArthur, 2024).

Thank you for this comment.

Methods

- A point to consider walking only (and not other modes) might be that this represents a lower-bound accessibility level. The authors might consider emphasizing this.

Thank you for bringing this up. We mentioned this point in the methods section (see answer below) and also discussed it in the later part of the paper.

- A 180-minute threshold is a lot. Are there any references on observed travel times?

Thank you for your comment. We used the threshold, since it was also mentioned by Weiss et al. in their global study on accessibility. After a comprehensive discussion following your comment, we decided to limit ourselves to 120 min, to also be able to compare our results with national wide analyses (e.g. Ouma et al. 2018).

- From the manuscript, it is not clear accessibility to whom? The standard 5 kph speed suggests it is a healthy population. What are the implications of this in the findings? In any case, further discussion is needed based on the evidence in the literature.

Thank you for this comment. This point was also brought up by other reviewers. We considered it and reran the analysis with a lower walking speed. Besides, we added several studies to explain the choices in more detail

“The mode of travel was defined as walking, as it is reported to be the main mode of transportation in SSA \cite{Sietchiping.2012} and is independent of income. Additionally, it is reported that for example in Kenya more than 80% of patients access healthcare by walking \cite{watmough2022using} and the corresponding travel time can be seen as lower-bound. Similar to other studies analysing walking travel times to healthcare facilities \cite{watmough2022using}, we set the walking speed to 3.6 km/h. The OpenStreetMap (OSM) road network has no directional restrictions, allowing any segment to be traversed freely. However, because OSM primarily maps roads intended for motorized vehicles, it lacks data on footpaths, shortcuts, or informal paths that could reduce travel time between origins and destinations for pedestrians. Additionally, OSM data does not provide precise information on factors such as road classification, speed limits, sidewalk presence, road conditions or materials, barriers, or slopes, all of which could adversely impact walking speed. However, it is reported that pedestrians often walk along the side of main roads, so they are a good proxy for possible paths. To account for intra-urban variations in walking speed (e.g. due to topography) and to test the robustness of the results, we varied the speed by $\pm 20\%$ similar to Ouma et al. \cite{Ouma.2018} or Watmough et al. \cite{watmough2022using}.”

- Topography is demonstrated to be an important factor affecting walking time. How does this vary from city to city? How is this considered in the analysis? If it is not, measures of uncertainty would be needed.

Please see detailed response above.

- What are the parameters and rules used for walking routes? In developing context, the limited pedestrian infrastructure can make residents to rely on road infrastructure. How is this considered in the routing model?

Thank you for raising this. There are no directional restrictions on the road network and any segment of the network can be traversed. As the OSM data covers roads for motorized vehicles, there is no systematic information on footpaths, shortcuts or trodden paths that could potentially reduce time in the origin-destination matrix. On the other hand, there is inconsistent information on road class, speed limits, presence of sidewalks, road condition/material, barriers and slopes that could potentially negatively influence walking speed. It is of course not unlikely that a motorized mode of transportation is chosen before walking in case of need for hospitalization if possible. In any case, as our analysis is distance-based, the (dis)advantages of distances from home to healthcare units are independent of transportation mode.

We revised the description of the road network in the methods section as follows:

“The mode of travel was defined as walking, as it is reported to be the main mode of transportation in SSA \cite{Sietchiping.2012} and is independent of income. Additionally, it is reported that for example in Kenya more than 80\% of patients access healthcare by walking \cite{watmough2022using} and the corresponding travel time can be seen as lower-bound. Similar to other studies analysing walking travel times to healthcare facilities \cite{watmough2022using}, we set the walking speed to 3.6 km/h. The OSM road network allows free traversal but mainly maps motorized roads, lacking systematic data on footpaths or shortcuts that could reduce pedestrian travel time. Additionally, OSM data doesn't consistently detail road classifications, conditions, or other factors affecting walking speed, but main roads are often used as proxies for pedestrian paths \cite{watmough2022using}.

- It would be important to briefly discuss the key aspects of the healthcare data in more detail. This is a crucial input of the analysis and would increase the credibility and robustness, e.g. collection methods, validation, and coverage, time period.

Thank you for this important comment. We added the following paragraph to provide more information on the health care data used in our analysis.

“We use data on health-care facilities from Maina et al. \cite{Maina.2019}. They collected information on 96395 geocoded health facilities in SSA until 2018 by using data from ministries of health, state agencies, United Nations Coordination of Humanitarian Affairs and other sources. Further information on the data set can be found in the data descriptor paper.”

- Additionally, the authors mention that they have “categorized them into two groups according to the table provided as supplementary material. While the first group (i) refers to hospitals and centres with surgical care, the second group (ii) includes health clinics and other small health facilities.” This should be elaborated further, what is the key criteria employed and what is the rationale behind?

Thank you for this comment. We rephrased the sentences in our methods section to be more clear in our methodology. The main reason for dividing the groups was to be able to compare our results with other studies and to give the readers the opportunity to distinguish between smaller and bigger health care facilities.

“We calculated service areas to healthcare facilities for all 19 cities. Additionally, we performed an analysis just focusing on hospitals to be able to compare the results of our methodology with similar studies \cite{Ouma.2018,banke2024geographical}. For detailed information on the classification of both groups of health facilities see below.”

- How does the size/capacity of the facility might influence the results? This is partially acknowledged in the limitations. Can the authors elaborate how is this expected to affect the results? Age of the population can play a role.

Thank you for bringing this up. We added a short paragraph to reflect on this point:

“Another possible bias in the dataset is that information on healthcare facilities could be outdated (e.g., due to destruction in armed conflicts). Additionally, the capacity of healthcare facilities was omitted from our analysis, limiting our ability to conduct a more nuanced evaluation. Larger facilities with greater capacity typically serve more people, potentially skewing accessibility results if not considered, as they may reduce pressure on smaller, nearby facilities. Conversely, smaller facilities might be overwhelmed, reducing access and increasing wait times, especially in densely populated areas. Incorporating capacity considerations would have enabled a more detailed analysis, such as a two-step floating catchment area approach.

Moreover, the age of the population plays a crucial role, as older populations generally have higher healthcare needs and more frequent service requirements. Areas with older demographics might experience greater strain on local facilities, complicating accessibility further. Future studies should incorporate both facility capacity and population age to enhance the accuracy of healthcare accessibility assessments.”

- What are the considerations taken to assess/trust the completeness of the OSM road network data? This might bias the findings substantially. Are there references for the cases studied? This has been done for building data in (Herfort et al., 2023).

Thank you for raising the important point regarding the completeness of the OSM road network data. While we acknowledge that OSM data can vary in accuracy and completeness, especially in developing regions, previous studies have found OSM to be a robust source for urban analysis due to its frequent updates and extensive community validation efforts, particularly for humanitarian applications (<https://www.hotosm.org/>). Our decision to use OSM data is based on literature recommendations and previous experience with OSM datasets in an African context. For instance, in [1], OSM data street network data were used to recreate city blocks successfully in Sub-Saharan African cities. While the OSM data are not perfect, particularly in some areas of the Global South, its completeness is rapidly increasing. Additional recommendations from global analyses support this statement [2,3]. We do agree that a comparison with reference data would have been ideal but such datasets were unavailable for the case studies. We preferred to use OSM data rather than the for instance, the Microsoft Road Network product (<https://github.com/microsoft/RoadDetections>), as it is a user-driven manually dynamically evolving product and is not based on modelled outputs (which can induce even more uncertainty (especially in informal settlements where the road network is not typically drawn from training data that Microsoft/Google models are trained with. Once these datasets become ingested to the OSM database (which would imply passing quality checks), we will definitely include them. Finally, we also aim to use this study as a first benchmark for similar work, and we are confident that OSM allows for better replication as they are continuously updated, while the products available from Microsoft or Google are typically only available for a single time stamp.

However, we recognize that any gaps in the OSM road network could introduce some bias into our findings. As such, our findings should be interpreted with these potential limitations in mind. Future research could benefit from more detailed assessments of OSM data completeness, similar to those done for building data [4]. We have included these aspects in the revised section of the manuscript.

„We recognize that the potential incompleteness of OSM data could introduce a degree of bias into our findings. However, OSM street network data have been successfully used to replicate land-use parcels \cite{grippa2018mapping}, or improving land cover products \cite{fonte2017generating} in several African cities. Moreover, global analyses are quite encouraging and highlight the constant improvement of data quality and completeness \cite{barrington2017world, barrington2019global}. Future studies should consider evaluating the road completeness as was recently done by Herfort \cite{Herfort.2023} for building footprints.”

References

[1] Grippa, T., Georganos, S., Zarougui, S., Bognounou, P., Diboulo, E., Forget, Y., Lennert, M., Vanhuyse, S., Mboga, N. and Wolff, E., 2018. Mapping urban land use at street block level using openstreetmap, remote sensing data, and spatial metrics. *ISPRS International Journal of Geo-Information*, 7(7), p.246.

[2] Barrington-Leigh, Christopher, and Adam Millard-Ball. "The world's user-generated road map is more than 80% complete." *PloS one* 12.8 (2017): e0180698.

[3] Barrington-Leigh, Christopher, and Adam Millard-Ball. "A global assessment of street-network sprawl." *PloS one* 14.11 (2019): e0223078.

[4] Herfort, B., Lautenbach, S., Porto de Albuquerque, J., Anderson, J. and Zipf, A., 2023. A spatio-temporal analysis investigating completeness and inequalities of global urban building data in OpenStreetMap. *Nature Communications*, 14(1), p.3985.

• Could the authors clarify the rationale and interpretation of the relative risk in lines 168 to 169? Thank you for bringing this up. We wrote two additional sentences to explain the rationale and interpretation of this indicator in more detail:

“By setting the SA for both land use types in relation, we calculate the relative risk $RR = SA_{\text{Formal}} / SA_{\text{Informal}}$ (Formal SA and Informal SA) for different steps in time. The relative risk (RR) quantifies the disparity in healthcare accessibility between formal and informal areas. An RR greater than 1 indicates higher accessibility in formal areas, while an RR less than 1 suggests greater accessibility in informal areas.”

Results

• Section 3.1 could re-iterate the type of health-care facilities included in to improve clarity for readers.

• Figure 1, can the authors clarify the interpretation of error bars? We added information on the error bars in the caption of Figure 1.

• Figure 2 is hard to read; could it use annotation? Also, removing error shade can improve clarity. Also, the authors are encouraged to consider using an accessible blind-colour palette.

Thank you for your comment. We removed the error bars in Figure 2 to make it easier to read.

• The header of section 3.2 should reflect that this is about residential land use only (locations of hospitals is land use too).

Thank you for this comment. We corrected the header of section 3.2.

• Figure 3, adding column and rows labels would help to increase clarity.

Thank you for this comment. Unfortunately, we don't under

• In figure 3, top-right panel, there is a big gap between the three cities with lowest accessibility and all of the rest. Is there an explanation from authors?

• Table 2 and 3, Including the meaning of intervals in square brackets in a footnote or caption can improve readability.

Thank you for this comment. We included information on the uncertainty into the captions of the tables.

Discussion

• The authors are claiming to confirm the importance of medium-size cities. But it is not very clear how these type of cities play a relevant role based on the results. Could the authors clarify?

Good point. Our findings suggest that medium-sized cities may lack the infrastructure development or resource allocation seen in larger cities, leading to more pronounced inequalities. From our interpretation and based on the literature this can be explained by the disproportionate focus on provisions for the largest metropolitan centers. As stated by the UN 2016 report “ *The fastest growing urban centres are the small and medium cities with less than one million inhabitants, which account for 59 per cent of the world’s urban population and 63 per cent of the urban population in Africa. Despite the demographic importance and potential role of such cities, urban planning efforts in developing countries have focused disproportionately on the problems of large metropolitan areas, thereby contributing to urban primacy. If small and medium cities are to fulfil their potential, then they should form part of the new urban agenda for developing countries*”. The sometimes much higher urban growth rate of medium sized cities in comparison to primary cities has been also been confirmed by Zimmer et al. [2]. Consequently, we can argue that the high growth cities of medium sized cities has not been aligned with provisions to accessibility as in primary cities (or potential small cities where the population is not growing as fast). In the revised manuscript we include the following:

“For the city sample we investigated in this study, we therefore find, that individuals in informal areas in medium-sized cities are disadvantaged when it comes to accessibility to health-care facilities. This finding can be explained by the mismatch in medium sized cities between rapid urban growth (often exceeding primary cities) and provision of resources in infrastructure and healthcare (Zimmer et al, UN Habitat report)

References

- [1] Un-Habitat, 2016. *World Cities Report 2016: Urbanization and development-emerging futures*. UN.
- [2] Zimmer, A., Guido, Z., Tuholske, C., Pakalniskis, A., Lopus, S., Caylor, K. and Evans, T., 2020. *Dynamics of population growth in secondary cities across southern Africa*. *Landscape Ecology*, 35, pp.2501-2516.

Reviewer #4 (Remarks to the Author):

The authors present an interesting approach to a very complicated issue. We have known for a long time that patterns of spatial accessibility in urban areas are very different from those in rural areas, but addressing this problem in research remains a challenge to this day because many traditional approaches, such as the raster-based approach, are unfit for urban areas. The authors have used a methodology to assess this complicated issue. The paper is interesting to read and addresses an important issue. The authors have thought about many of the assumptions that feed into this methodology, such as “how do you define a city?” and “how do you separate informal areas and formal areas?”. However, in my opinion, some important challenges still need to be addressed to make the paper more robust and publishable.

Major concerns:

1) The authors have used the network analyst approach. This approach needs a continuous/connected road network without interruptions or missing segments. In my opinion, this is especially hard to achieve in informal settlements where OpenStreetMap hasn't been able to digitize all road segments yet. The authors have not mentioned how they ensured that all road segments are connected and what they did to minimize bias in missing road segments in general and particularly in informal settlements? Have they considered the AI digitized road segments from data for Good from Meta? (<https://github.com/facebookmicrosites/Open-Mapping-At-Facebook/wiki/Available-Countries>)

Thank you for raising the important point regarding the completeness of the OSM road network data. While we acknowledge that OSM data can vary in accuracy and completeness, especially in developing regions, previous studies have found OSM to be a robust source for urban analysis due to its frequent updates and extensive community validation efforts, particularly for humanitarian applications (<https://www.hotosm.org/>). Our decision to use OSM data is based on literature recommendations and previous experience with OSM datasets in an African context. For instance, in [1], OSM data street network data were used to recreate city blocks successfully Sub-Saharan African cities. While the OSM data are not perfect, particularly in some areas of the Global South, its completeness is rapidly increasing. Additional recommendations from global analyses support this statement [2,3]. We do agree that a comparison with reference data would have been ideal but such datasets were unavailable for the case studies. We preferred to use OSM data rather than the for instance, the Microsoft Road Network product (<https://github.com/microsoft/RoadDetections>), as it is a user-driven manually dynamically evolving product and is not based on modelled outputs (which can induce even more uncertainty (especially in informal settlements where the road network is not typically drawn from training data that Microsoft/Google models are trained with). Once these datasets become ingested to the OSM database (which would imply passing quality checks), we will definitely include them. Finally, we also aim to use this study as a first benchmark for similar work, and we are confident that OSM allows for better replication as they are continuously updated, while the products available from Microsoft or Google are typically only available for a single time stamp.

However, we recognize that any gaps in the OSM road network could introduce some bias into our findings. As such, our findings should be interpreted with these potential limitations in mind. Future research could benefit from more detailed assessments of OSM data completeness, similar to those done for building data [4]. We have included these aspects in the revised section of the manuscript.

“We recognize that the potential incompleteness of OSM data could introduce a degree of bias into our findings. However, OSM street network data have been successfully used to replicate land-use parcels (Grippa et al), or improving land cover products (Fonte et al.) in several African cities. Moreover, global analyses are quite encouraging and highlight the constant improvement of data quality and completeness [cite{barrington2017world, barrington2019global}]. Future studies should consider evaluating the road completeness as was recently done by Herfort [cite{Herfort.2023}] for building footprints.”

References

[1] Grippa, T., Georganos, S., Zarougui, S., Bognounou, P., Diboulo, E., Forget, Y., Lennert, M., Vanhuyse, S., Mboga, N. and Wolff, E., 2018. Mapping urban land use at street block level using openstreetmap, remote sensing data, and spatial metrics. *ISPRS International Journal of Geo-Information*, 7(7), p.246.

[2] Barrington-Leigh, Christopher, and Adam Millard-Ball. "The world's user-generated road map is more than 80% complete." *PloS one* 12.8 (2017): e0180698.

[3] Barrington-Leigh, Christopher, and Adam Millard-Ball. "A global assessment of street-network sprawl." *PloS one* 14.11 (2019): e0223078.

[4] Herfort, B., Lautenbach, S., Porto de Albuquerque, J., Anderson, J. and Zipf, A., 2023. A spatio-temporal analysis investigating completeness and inequalities of global urban building data in OpenStreetMap. *Nature Communications*, 14(1), p.3985.

2) The authors have chosen a walking speed of 5km/h which is quite high in the case of people needing hospital care. How did the authors come up with this speed and on which target population was it based? I would suggest the authors to lower this speed and also have a look at the following paper that discusses walking speeds for mothers and children in more depth: Watmough, G.R., Hagdorn, M., Brumhead, J. et al. Using open-source data to construct 20 metre resolution maps of children's travel time to the nearest health facility. *Sci Data* 9, 217 (2022). <https://doi.org/10.1038/s41597-022-01274-w>.

Thank you for this valuable input. We were not aware of these references and incorporated them into our analyses. We reduced the walking speed to 3.6 km/h and reran the analysis using this adjusted walking speed.

"The mode of travel was defined as walking, as it is reported to be the main mode of transportation in SSA \cite{Sietchiping.2012} and is independent of income. Additionally, it is reported that for example in Kenya more than 80% of patients access healthcare by walking \cite{watmough2022using} and the corresponding travel time can be seen as lower-bound. Similar to other studies analyzing walking travel times to healthcare facilities \cite{watmough2022using}, we set the walking speed to 3.6 km/h. The OSM road network allows free traversal but mainly maps motorized roads, lacking data on footpaths or shortcuts that could reduce pedestrian travel time. Additionally, OSM data doesn't detail road classifications, conditions, or other factors affecting walking speed, but main roads are often used as proxies for pedestrian paths \cite{watmough2022using}. To account for intra-urban variations in walking speed (e.g. due to topography) and to test the robustness of the results, we varied the speed by +/-20% similar to Ouma et al. \cite{Ouma.2018} or Watmough et al. \cite{watmough2022using}."

3) The authors make quite large assumptions on the classification of specific types of health facilities into two groups and say that one of the groups has surgical capacity. Unfortunately, many of these assumptions have proven to be impossible because types of health facilities are rarely a good indication of the types of services they provide, see (Petragallo S, Timoner P, Hierink F, et al Assessing the accuracy of health facility typology in representing the availability of health services: a case study in Mali. *BMJ Open* 2024;14:e077127. doi: 10.1136/bmjopen-2023-077127). What is the main purpose of dividing these facilities into these groups? I think that the overall statistics are already interesting in itself without the sub-classification.

Thank you for this comment. We rephrased the sentences in our methods section to be more clear in our methodology. The main reason for dividing the groups was to be able to compare our results with other studies and to give the readers the opportunity to distinguish between smaller and bigger health care facilities.

“We calculated service areas to healthcare facilities for all 19 cities. Additionally, we performed an analysis just focusing on hospitals to be able to compare the results of our methodology with similar studies \cite{Ouma.2018,banke2024geographical}. For detailed information on the classification of both groups of health facilities see below.”

Thereby, we shifted our focus in our study away from the two groups of health facilities. We did therefore not include the reference you mentioned.

4) In several places in the manuscript the authors compare their results to previous studies, such as Weiss et al. and Ouma et al. However, the raster-based approach in these studies also includes rural areas and the approach also allows speeds for other types of landcovers. In addition, at the time of these studies only the unconstrained data was available for the gridded population, meaning that a comparison in the statistics is complicated. I would highly recommend the authors to look at papers that assessed access to healthcare specifically in urban areas and using different assessment techniques, such as the work recently published by Banke-Thomas, Aduragbemi, et al. "Geographical accessibility to functional emergency obstetric care facilities in urban Nigeria using closer-to-reality travel time estimates: a population-based spatial analysis." The Lancet Global Health 12.5 (2024): e848-e858. How does their approach and findings relate to these results, are their coverage stats higher/lower and why is that?

We carefully used these and other additional references to update our introduction to address this point.

“Banke-Thomas et al. \cite{banke2024geographical} took a first step towards intra-urban analysis of access to maternity care in SSA by analyzing travel times for 15 Nigerian cities. They found that informal settlements appear to be worse off compared to parts of the inner city. Dumedah et al. \cite{dumedah2023inequities} analyzed the access to healthcare in 4 Ghanaian cities. Similar to the studies mentioned above, they find that relative wealth negatively correlates with access to healthcare in the cities investigated.”

As suggested, we also compared our results with the study you mentioned. Here, like in the comparison above, it is important to not compare similar values. Again, like in the national comparison, we just focused on public facilities and compared the accessibility values for two cities (Lagos and Ibadan). We found significantly lower values for both cities, when using our approach. Although the number of facilities is the same, the mode of transport makes a big difference. We tried to explain the differences as good as possible, highlighting the fact, that our approach with focusing only on walking is more conservative, than the approach of Banke-Thomas et al. We added the following paragraphs to our paper.

“In our study, we analyzed accessibility to healthcare facilities specifically by walking, whereas Banke-Thomas et al. \cite{banke2024geographical} assessed accessibility to comprehensive emergency obstetric care facilities across 15 Nigerian cities using travel time estimates from Google Maps, which primarily include driving as the mode of transportation. This methodological difference allows for a meaningful comparison between the two approaches in the context of Lagos and Ibadan.

For Ibadan, Banke-Thomas et al. \cite{banke2024geographical} reported spatial accessibility (SA) of 44.9\% within 15 minutes and 79.8\% within 30 minutes when considering only public facilities. In contrast, our study, which focuses solely on hospitals (Group (i)) and considers walking as the mode of travel, found significantly lower SA values of 2.9\% for 15 minutes and 12.6\% for 30 minutes. Similarly, in Lagos, they found SA values of 45.8\% for 15 minutes and 83.7\% for 30 minutes, while our study found 4\% (15 min) and 15.9\% (30 min).

These stark differences in SA percentages can be primarily attributed to the mode of transportation considered in each study. By focusing on walking, our approach is inherently more conservative and is designed to detect population groups with insufficient access to healthcare under more constrained mobility conditions. In contrast, Banke-Thomas et al.'s \cite{banke2024geographical} use of driving times likely reflects faster travel times and greater accessibility in urban areas. Although the number of hospitals considered was similar between the two studies, the choice of transportation mode plays a crucial role in accessibility outcomes. Therefore, while both sets of results are reasonable within their respective contexts, our conservative approach highlights potential disparities in healthcare access that might be overlooked when considering faster modes of transportation."

5) The authors used WorldPop constrained UN adjusted data. I would recommend them to mention the uncertainty of estimating population in informal settlements in the limitation section, as mentioned in these two articles:

- Thomson, D.R.; Gaughan, A.E.; Stevens, F.R.; Yetman, G.; Elias, P.; Chen, R. Evaluating the Accuracy of Gridded Population Estimates in Slums: A Case Study in Nigeria and Kenya. *Urban Sci.* 2021, 5, 48. <https://doi.org/10.3390/urbansci5020048>
- Breuer, Julius HP, et al. "The unseen population: Do we underestimate slum dwellers in cities of the Global South?." *Habitat International* 148 (2024): 103056.

Thank you for this comment. We are aware of the uncertainties of gridded population data sets and added two extra paragraphs (one in the methods section and one in the limitations section) to emphasize the importance of this aspect.

"Although the data sets used here can be highly uncertain, especially in the area of informal settlements (cf. \cite{thomson2021evaluating, breuer2024unseen}), they are nevertheless the best data sets available on a continental scale and thus make comparative studies possible in the first place."

"It is acknowledged that gridded population data sets, such as the WorldPop data set that was utilized in the analysis, exhibit considerable uncertainty when estimating the number of individuals residing in slums. A comparison of gridded population estimates with demographic health survey data \cite{thomson2021evaluating} or other data from the literature \cite{breuer2024unseen} in different cities in the Global South indicates that the number of individuals residing in informal areas may be ten times higher than suggested by the estimates. In comparing local survey data with gridded population data sets, Abascal et al. \cite{abascal2024making} demonstrated that self-reported population numbers are, on average, up to double the size of gridded population estimates. Consequently, the values calculated for spatial accessibility can be considered a lower boundary for the actual values."

Minor concerns:

1) Line 57: "SA is addressed as an indicator in the sustainable development goal (SDG) 3.8." it is implied but not specifically mentioned nor formulated as an indicator or measurement. Suggestion to reformulate.

Thank you for this comment. We were not specific enough in our first formulation. We rephrased the sentence in the following way:

"Spatial accessibility (SA) to health-care services is an important factor when it comes to human well-being. When target 3.8 of the United Nations Sustainable Development Goals calls for universal health coverage, SA is a necessary prerequisite."

2) Line 80-81, there are increasing publications on urban access to care in SSA. See articles:

- a. Macharia, Peter M., et al. "A geospatial database of close-to-reality travel times to obstetric

emergency care in 15 Nigerian conurbations." *Scientific data* 10.1 (2023): 736.

b. Geographical accessibility to functional emergency obstetric care facilities in urban Nigeria using closer-to-reality travel time estimates: a population-based spatial analysis." *The Lancet Global Health* 12.5 (2024): e848-e858.

We carefully used these and other additional references to update our introduction to address this point.

"Banke-Thomas et al. \cite{banke2024geographical} took a first step towards intra-urban analysis of access to maternity care in SSA by analyzing travel times for 15 Nigerian cities. They found that informal settlements appear to be worse off compared to parts of the inner city.

Dumedah et al. \cite{dumedah2023inequities} analyzed the access to healthcare in 4 Ghanaian cities. Similar to the studies mentioned above, they find that relative wealth negatively correlates with access to healthcare in the cities investigated."

3) Line 108-109: "How good is the walking accessibility to health-care facilities for different cities in SSA and how does this compare to global or national estimates?" I would recommend the authors to reduce this comparison to other comparable studies that have looked at urban contexts or used a similar method as indicated in the major concerns.

Thank you for this important comment. Please see our response to your comment No. 4.

4) Line 169: RR = SAF/SAIF, can the others at the abbreviation for "F" and "IF" above? Such as Formal (F) and Informal (IF)?

Thank you for this remark. We reformulated the text.

5) Table 1, last column. If only counting the number of hospitals, I would suggest renaming the column to number of hospitals instead of health facilities (just hospitals).

Thank you very much for this comment. We renamed the column to make the distinction clearer:

"Number of health care facilities / thereof hospitals"

6) Line 258-261: You mean the second closest within 15 minutes of the first closest? Please rephrase.

Thank you for this comment. We updated the text.

7) Caption for figure two: I think it would be easier to mention the health facilities in group 2 e.g. clinics and smaller health facilities + mention whether it is in full urban areas or just in formal or informal settlements.

Thank you for this comment. We adjusted the figure and the caption, to make the content clearer.

8) Line 312: you miss the word "than", significantly higher "than" 1.

Thank you, we corrected this error.

Dear Katherina Barnes,

Thank you and the reviewer for your positive and useful feedback on our paper entitled "Differences in walking access to healthcare facilities between formal and informal areas in 19 sub-Saharan African cities".

We revised the manuscript according to the short comments, the reviewers provided:

To address the two comments Reviewer #2 made, we added a more detailed explanation on the buffer we used and added a sentence to explain the selection of study sites.

We addressed the comments of Reviewer #3 by annotating some curves and referring to the Figure in the appendix for full description. We also checked we the color palette used and the company providing the color palette told us, that it is accessible for color blind readers.

We also completed the final submission file checklist and created the Zenodo repository including all the data and codes necessary to perform the analyses and create the figures.

All changes are highlighted in the revised manuscript - marked up.

If any questions remain open, please do not hesitate to contact us.

Best regards,

The authors